# Succinate and its G-protein-coupled receptor stimulates osteoclastogenesis

Yuqi Guo[1], Chengzhi Xie[1,†], Xiyan Li[2], Jian Yang[1], Tao Yu[1,3], Ruohan Zhang[1], Tianqing Zhang[1], Deepak Saxena[1], Michael Snyder[2], Yingjie Wu[3,4] & Xin Li[1,5,6]

The mechanism underlying bone impairment in patients with diabetes mellitus, a metabolic disorder characterized by chronic hyperglycaemia and dysregulation in metabolism, is unclear. Here we show the difference in the metabolomics of bone marrow stromal cells (BMSCs) derived from hyperglycaemic (type 2 diabetes mellitus, T2D) and normoglycaemic mice. One hundred and forty-two metabolites are substantially regulated in BMSCs from T2D mice, with the tricarboxylic acid (TCA) cycle being one of the primary metabolic pathways impaired by hyperglycaemia. Importantly, succinate, an intermediate metabolite in the TCA cycle, is increased by 24-fold in BMSCs from T2D mice. Succinate functions as an extracellular ligand through binding to its specific receptor on osteoclastic lineage cells and stimulates osteoclastogenesis *in vitro* and *in vivo*. Strategies targeting the receptor activation inhibit osteoclastogenesis. This study reveals a metabolite-mediated mechanism of osteoclastogenesis modulation that contributes to bone dysregulation in metabolic disorders.

[1] Department of Basic Science and Craniofacial Biology, New York University College of Dentistry, New York, New York 10010, USA. [2] Department of Genetics, Stanford University, Stanford, California 94305-5120, USA. [3] Institute for Genomic Engineered Animal Models of Human Diseases, Liaoning 116044, China. [4] Advanced Institute for Medical Science, Dalian Medical University, 9 West Section, South Lvshun Road Dalian, Liaoning 116044, China. [5] Department of Urology, New York University Langone Medical Center, New York, New York 10016, USA. [6] Perlmutter Cancer Institute, New York University Langone Medical Center, New York, New York 10016, USA. † Present address: Department of Pediatrics, Duke University School of Medicine, Durham, North Carolina 27710, USA. Correspondence and requests for materials should be addressed to Xin L. (email: xl15@nyu.edu).

Bone fragility has emerged as a complication of diabetes mellitus, a metabolic disorder characterized by chronic hyperglycaemia with dysregulation in metabolites. More than 90% of diabetes mellitus cases are type 2 diabetes (T2D), which is associated with insulin resistance in peripheral tissues. T2D is associated with trabecular defects and increases in the cortical porosity of bone[1]. However, the pathophysiological mechanism underlying diabetes and compromised bone metabolism is not clear. In terms of the cellular effects, whether there is a direct and causative link between abnormal metabolites resulting from hyperglycaemia and impaired cell functions in bone maintenance is unknown.

As a highly dynamic tissue, bone undergoes constant remodelling that maintains skeletal homoeostasis via a balance between bone resorption and bone formation. Osteoporosis occurs when the removal of old bone outpaces the formation of new bone[2–4]. Osteoclasts (OC), which are differentiated from haematopoietic progenitor cells, are responsible for bone resorption. Overactive OCs lead to imbalanced bone remodelling commonly associated with metabolic bone diseases including periodontal disease, arthritis and osteoporosis[5,6]. Although the trabecular bone density is usually maintained in T2D patients, increased cortical porosity has been observed in T2D patients[7–9], which indicates an increased OC activity and bone loss. Overall, patients with T2D exhibit increased osteoblast apoptosis, diminished osteoblast differentiation and enhanced OC-mediated bone resorption[10]. Animal models of diabetes all demonstrate enhanced osteoclastogenesis and bone resorption[11–13]. Ironically, a high glucose concentration inhibits RANKL-induced osteoclastogenesis in vitro[14,15]. Therefore, the enhanced OC activity in diabetic animal models may not be a direct consequence of hyperglycaemia.

We reported a significant metabolomics alteration from diabetic mice at the organ level in bone marrow[16]. We found that succinate, an intermediate metabolite in the tricarboxylic acid (TCA) cycle, was abnormally accumulated in the bone marrow of diabetic mice, which echoes a previous report on increased succinate in proliferative diabetic retinopathy patients[17]. Succinate is generated from succinyl-CoA in the TCA cycle via succinyl-CoA ligase. Beyond its traditional function as a metabolite in the TCA cycle, succinate also has a hormone-like function through the activation of succinate receptor 1 (SUCNR1), a G-protein-coupled receptor[18]. SUCNR1 is highly expressed in mouse kidneys, livers, spleens and the small intestine[18]. Hyperglycaemia causes succinate accumulation and SUCNR1 activation in retinal ganglion cells in streptozotocin (STZ) -induced diabetes rats[19]. Succinate-activated SUCNR1 induces the expression of pro-inflammatory cytokines in a hypoxia-inducible factor-1 alpha (HIF-1α)-independent manner[20]. Specifically, succinate stimulation of SUCNR1 enables dendritic cell-mediated T-cell activation[21], which is able to induce osteoclastogenesis[22]. Therefore, we hypothesize that elevated succinate may enhance osteoclastogenesis and bone loss via SUCNR1 activation.

In this study, we use friend virus B (FVB) wild-type (WT) and MCK-KR-hIGF-IR (MKR) transgenic mice as normal and hyperglycaemia models, respectively. Due to the tissue-specific expression of a dominant-negative mutant of human IGFI receptor that interferes with glucose uptake in muscles, MKR mice rapidly develop severe diabetes within 2 months[23]. MKR mice have slender bones and exhibit skeletal fragility and susceptibility to fracture due to reduced transverse bone accrual and increased osteoclastogenesis[13]. Using in vivo and in vitro models, our study reveals an altered metabolite profile in bone marrow stromal cells (BMSCs) from T2D mice and demonstrates the function of stromal cell-derived succinate

through SUCNR1 and stimulation of NF-κB signalling to enhance osteoclastogenesis. Our findings have implications for the treatment of collateral bone damage in patients with diabetes.

## Results

**Abnormal succinate accumulation in hyperglycaemic conditions.** Recently, we reported the altered metabolome and less-efficient respiration activity in diabetic mouse bone marrow were demonstrated by the imbalances in many metabolites in the TCA cycle at the organ level[16]. To dissect the specific impacts of hyperglycaemia on the bone metabolism at the cellular level, we applied the same metabolomics approach to investigate the metabolite profiles in BMSC samples derived from normal (WT) and hyperglycaemic (MKR) male mice using liquid chromatography-mass spectrometry (LC-MS) -based metabolomics (Fig. 1a,b). Male MKR mice become severe diabetic at 12-week-old according to glucose tolerance test after overnight fasting (Supplementary Fig. 1a). We extracted 14,062 mass features (each defined by a pair of retention time and accurate mass) from the positive mode and 5,959 mass features from the negative mode. The raw data have been deposited into the Metabolomics Data Repository and Coordinating Center. Compared with WT BMSCs, 142 metabolites were significantly changed by >1.5-fold in MKR BMSCs; 126 were upregulated and 16 were downregulated (Supplementary Table 1). Since succinate is the first metabolite in the TCA cycle exhibiting abnormal accumulation in BMSCs due to diabetes, it might be a key metabolic factor responsive to the hyperglycaemia. In contrast to the barely detectable serum succinate level in WT mice, the serum succinate level in MKR mice was significantly elevated by more than 20-fold (Fig. 1c). Interestingly, the succinate level started to rise in 6–8-week-old MKR mice when the mice became diabetic and became significant when MKR mice reached 12-week-old (Supplementary Fig. 1b,c). Taken together with the fact that an abnormal accumulation of succinate level was observed in total bone marrow aspirate[16], T2D mice failed to control succinate in both global and local levels intracellularly and extracellularly. Importantly, high glucose culture significantly elevated succinate levels in human BMSCs (Fig. 1d). Succinate is converted to fumarate by succinate dehydrogenase (SDH) in the TCA cycle, and a deficiency in SDH activity could lead to the accumulation of succinate in the mitochondria, cytosol and eventually in the extracellular environment. We observed that SDH activity was drastically reduced by high glucose levels in both human and mouse BMSCs (Fig. 1e,f) which may explain the abnormal accumulation of succinate in the bone marrow. Meanwhile, the trabecular bone mass was reduced in MKR mice according to micro-computed tomography (μCT) analysis (Fig. 1g,h). The surface of OCs per bone surface based on tartrate resistance acid phosphatase (TRAP) staining (Fig. 1i,j) was greater in MKR mice than WT mice. The overall elevated OC activity was also reflected by the increase in the serum bone resorption marker TRAP5b levels (Fig. 1k). Consistent to previous reports, MKR mice are lean diabetic model with reduced body weight than the age-paired WT mice (Fig. 1l). The concurrence of elevated succinate and compromised bone phenotype in MKR mice indicates a regulatory role of succinate in bone metabolism. However, it is not clear whether a causal connection exists between succinate elevation and diabetes-related damages in bone.

**Succinate stimulates osteoclastogenesis and bone resorption.** Administration of succinate significantly increased OC numbers in bone marrow cell cultures in the presence of RANKL and M-CSF (Fig. 2) and RAW 264.7 cells, a homogenous cell population of macrophage lineage in the presence of RANKL

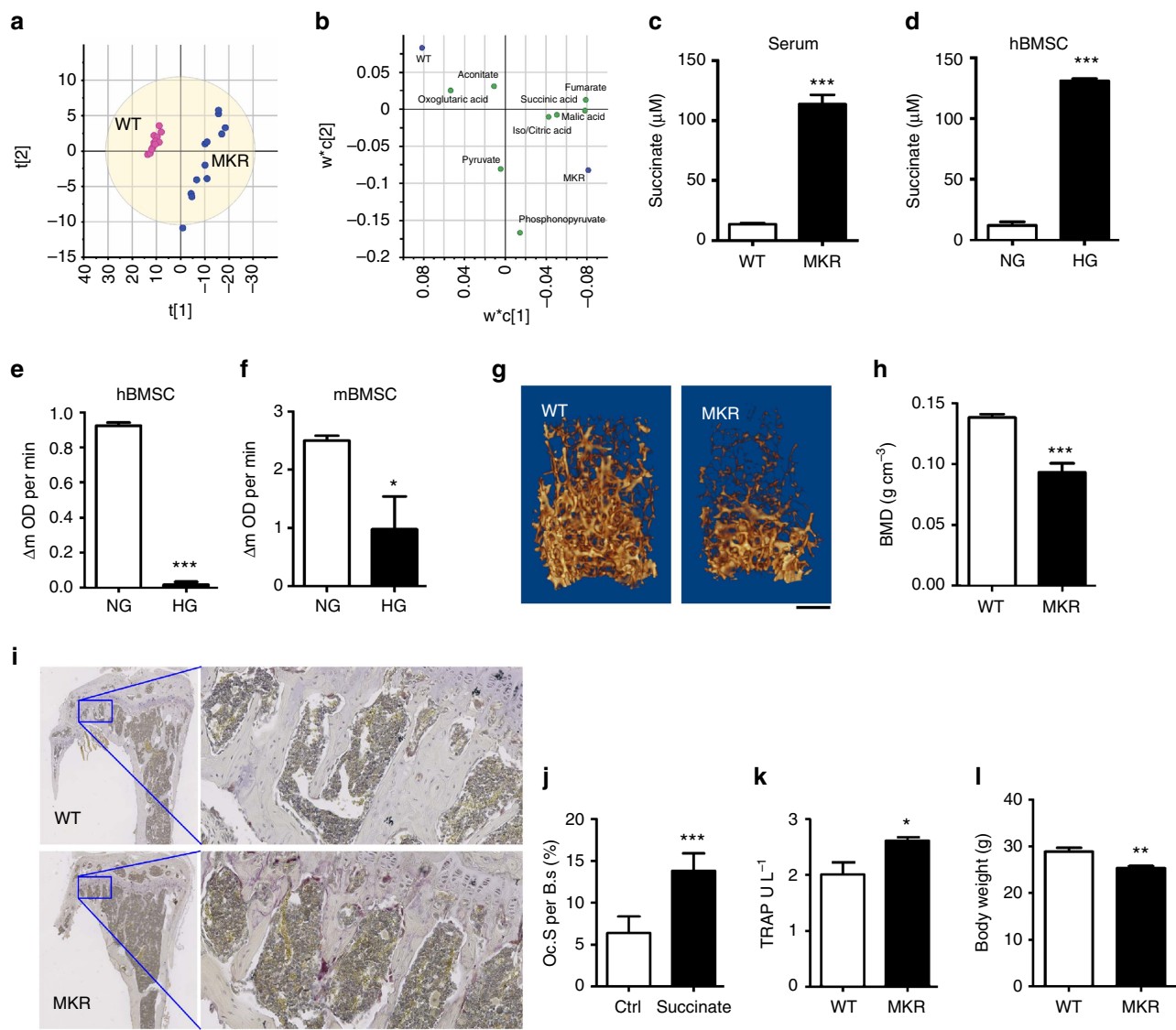

**Figure 1 | Hyperglycaemia increases succinate levels in BMSCs.** (**a**) A clustering scatter plot of 345 metabolites from cultured BMSCs of WT and MKR mice. Each circle represents one of three technical replicates from each of the four biological samples. The clustering used the PLS-DA model and unit-variance scaling in SIMCA (Umetrics, Sweden). R2Y = 0.997. Q2 = 0.971. The ellipse represents the 95% confidence interval. (**b**) The scatter plot of metabolite contribution to clustering shown in **a**. Only the metabolites from the TCA cycle (KEGG pathway 00020) detected in this study are shown for clarity. Each green circle represents a metabolite; each blue circle represents a reference point for each sample group. This plot was generated in SIMCA and formatted in Adobe Illustrator. Succinate levels of (**c**) mouse serum from WT or MKR mice; (**d**) hBMSCs cultured with normal (glucose = 1 g l$^{-1}$, NG) or high (glucose = 4.5 g l$^{-1}$, HG) glucose medium. (**e,f**) SDH activity in human and mouse BMSCs cultured with NG or HG glucose medium. (**g**) Representative μCT images scale bar, 500 μm and (**h**) Bone mineral density of distal femur from WT and MKR mice. (**i,j**) Representative TRAP staining images and quantitative result of TRAP + cell surface per bone surface (Oc. S per B.S) of tibia metaphysis from WT and MKR mice. Scale bars in **i**, left 1 mm, right 100 μm (**k**) Serum TRAP levels. (**l**) Body weights. Data show mean ± s.e.m., *P < 0.05, **P < 0.01, ***P < 0.001, according to a two-tailed t-test, n = 4. hBMSCs, human BMSC.

(Supplementary Fig. 2a,b)[24,25] based on TRAP staining, a widely used marker of differentiated OCs[26]. Succinate increased OC numbers in bone marrow cell cultures when added either in the past 3 days (D4–6) or from the beginning of the culture (D0–6). OC cultures exposed to succinate for longer time exhibited more OCs (Fig. 2a,b). Succinate also enhanced OC differentiation in a dose-dependent manner (Fig. 2c,d) and increased the area of the resorption pits by threefold (Fig. 2e,f) without affecting cell proliferation in the precursor cells (Supplementary Fig. 2c). Succinate administration stimulated the expression of marker genes for OC differentiation and maturation[27–29], including nuclear factor of activated T-cells, cytoplasmic 1 (NFATc1),

TRAP, cathepsin K (Ctsk) and the calcitonin receptor (CalR) (Fig. 2g), which further supports the assertion that extracellular succinate stimulates osteoclastogenesis in vitro.

To evaluate the efficacy of succinate in stimulating OCs in vivo, WT FVB mice were injected with succinate (Fig. 3). The daily injections with succinate for 7 weeks led to significantly elevated serum succinate levels (Fig. 3a) and significant bone loss (Fig. 3b–g). Of note, the precursor OC population was not regulated by succinate (Fig. 3h); the surface of mature OCs per bone surface (Fig. 3i,j) increased as a result of succinate administration. In addition to direct stimulation of osteoclastogenesis, succinate administration downregulated

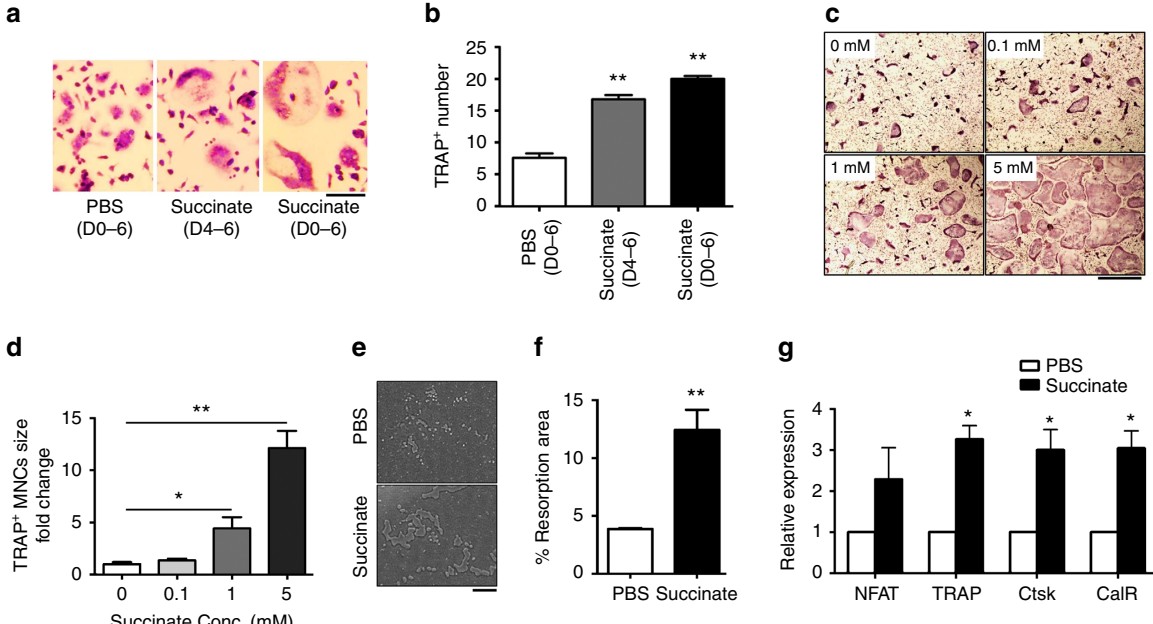

**Figure 2 | Succinate stimulates osteoclastogenesis *in vitro*.** As described in the method, the cells were seeded at 20,000 per well in 96-well plate and stimulated with the indicated concentration of cytokines (M-CSF, 30 ng ml$^{-1}$; RANKL, 50 ng ml$^{-1}$) from the time of culture; fresh cytokines were added every other day. (**a,b**) TRAP staining (scale bar, 100 μm) and enumeration of WT mouse OC cultures with PBS (D0 − 6), succinate administration for the last 3 days only (D4 − 6) or from the seeding (D0 − 6). (**c,d**) TRAP staining of OC cultures with succinate addition at the indicated concentrations. Scale bar, 500 μm. (**e,f**) Osteo Surface resorption assay with WT mice OC cell cultures treated with PBS or succinate (1 mM) for 6 days. Scale bar, 100 μm (**g**) OC differentiation marker gene expression by semi-quantitative real-time PCR from OC cell culture treated with PBS or succinate (1 mM) for 4 days. Data show mean ± s.e.m. (*n* = 5) *$P < 0.05$, **$P < 0.01$ according to a two-tailed *t*-test.

serum interleukin (IL)-4 and IL-13 levels (Fig. 3k) and upregulated osteogenic cytokine tumour-necrosis factor (TNF) and IL-1β levels (Fig. 3l,m). IL-4 and IL-13 are two closely related cytokines inhibiting OC formation[30,31]. Mice injected with succinate for 2 weeks exhibited a trend towards increases in OC number and activity (Supplementary Fig. 3) supporting the effects of succinate need be acquired cumulatively over time. The efficacy of succinate injection on bone resorption was also confirmed in WT C57/B6 mice (Supplementary Fig. 4), in which the 12-week-old male mice received succinate injection for 6 weeks. Consistent to the succinate effects in FVB mice, C57/B6 mice treated with succinate exhibited reduced bone mineral density, bone volume, trabecular number and increased trabecular separation according to μCT analysis (Supplementary Fig. 4a–f). Meanwhile, the serum resorption marker TRAP5b levels (Supplementary Fig. 4g) and osteogenic cytokine TNF and IL-1β levels (Supplementary Fig. 4h,i) were increased by succinate. Succinate administration did not alter the body weights of mice (Fig. 3n and Supplementary Fig. 4j).

**Targeting succinate signalling impedes osteoclastogenesis.** We examined the expression of the succinate-specific receptor SUCNR1 (ref. 18) in various bone cells. SUCNR1 exhibited an exclusively expression in haematopoietic lineage cells, as revealed by western blotting (Fig. 4a). In the stromal (BMSC) and osteoblast lineage cells (Pre-OB and OB), which exhibited elevated intracellular succinate levels in MKR mice (Supplementary Table 1), the expression of SUCNR1 was undetectable. In contrast, SUCNR1 was highly expressed in haematopoietic lineage cells such as mature murine OCs and their precursors (BM myeloid cells; Myeloid). This result suggests a non-autonomous mode of succinate action (that is, stromal cells released succinate functions through SUCNR1 on haematopoietic lineage cells in bone).

To determine the role of succinate and SUCNR1 in OCs, we synthesized a specific SUCNR1 antagonist 4c (refs 32–34) to inhibit SUCNR1 activation by succinate. We confirmed the structure of the synthetic compound using nuclear magnetic resonance spectroscopy (Supplementary Fig. 5a and high-performance liquid chromatography demonstrated a purity at 98.002% of this compound (Supplementary Fig. 5b). The impact of 4c at 0.1, 1, 10 and 50 μM in the OC culture derived from ficoll-processed mouse bone marrow cells were tested and 4c started to reduce the stimulation of succinate at 0.1 μM. At 1 μM it was able to completely erase the simulation of osteoclastogenesis by succinate *in vitro* (Fig. 4b,c). At higher concentration (50 μM), 4c inhibited osteoclastogenesis in both control and succinate-treated groups (Fig. 4b,c) which indicates a toxic effect of 4c at high dose. However, 4c at up to 100 μM has no inhibitory effect on the viability of pre-osteoclastic cells (Supplementary Fig. 5c). We further knockdown SUCNR1 with siRNA *in vitro*. Despite that the transfection with Lipofactamine-3,000 negatively affected cell attachment which left fewer cell numbers across the wells, succinate treatment still increased the number of TRAP-positive cells comparing to the control treated groups. Knocking down SUCNR1 with siRNA at 50 nM blunted the succinate stimulation on osteoclastogenesis (Supplementary Fig. 5d,e). The dependence of SUCNR1 for succinate effects in OCs is definitively supported by experiments with OCs derived from WT and SUCNR1 knockout (KO) mice. Succinate-enhanced osteoclastogenesis was abolished in bone marrow OCs derived from SUCNR1 KO mice (Fig. 4d,e). These results collectively suggest that succinate-enhanced osteoclastogenesis requires SUCNR1.

We observed that metformin re-set the altered metabolite balance in the bone marrow of diabetic mice[16]. Additionally, metformin treatment significantly reduced succinate elevations in BMSCs and sera from MKR mice (Fig. 5a,b and Supplementary Table 2). Metformin treatment regulated 81 metabolites in MKR

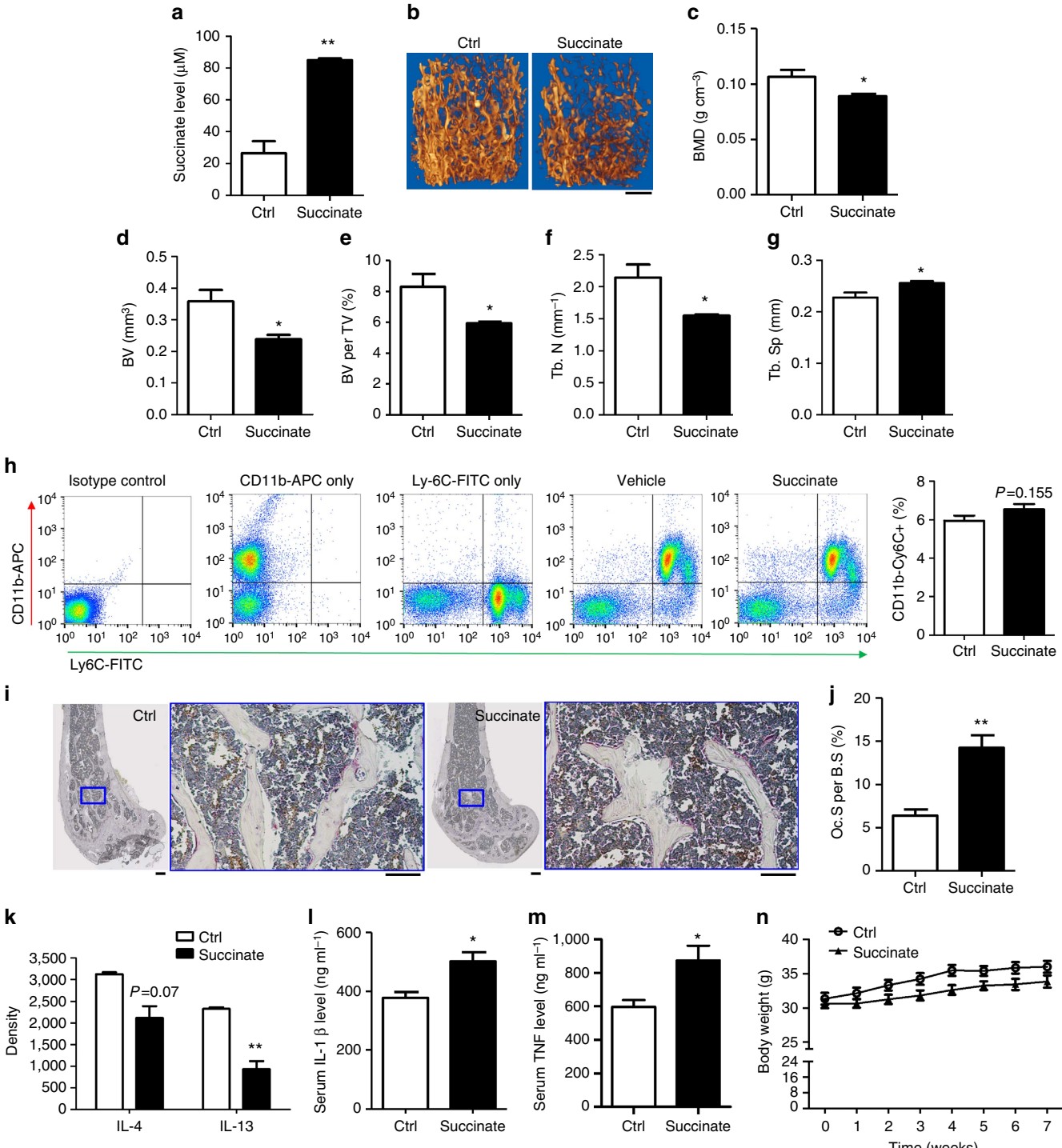

**Figure 3 | Succinate increases bone loss *in vivo*.** Three-month-old WT FVB mice were treated with PBS or a neutralized succinate solution for 7 weeks. (**a**) Serum succinate levels. (**b–g**) Representative µCT images and analysis with distal femur regions. Scale bar, 500 µm (**b**). (**h**) Representative flow cytometry plot showing gating and CD11b$^{-/lo}$Ly6C$^+$ populations in WT mice treated with PBS or succinate. The osteoclastic precursors population (CD11b$^-$Ly-6C$^+$) of bone marrow flush in are similar in the two treatment groups. (**i,j**) Representative TRAP staining images (scale bar, 250 µm) and quantitative result of TRAP$^+$ cell surface per bone surface (Oc. S per B.S). (**k–m**) Serum cytokine levels. (**n**) Body weights over 7 weeks. Data show mean ± s.e.m. ($n = 5$) *$P < 0.05$, **$P < 0.01$ by two-tailed $t$-test.

BMSCs; 14 metabolites were upregulated and 67 metabolites were downregulated (Supplementary Table 2). Interestingly, several key intermediate metabolites in the TCA cycle upregulated with hyperglycaemia—including succinate, fumarate and malate—were rebalanced by metformin (Supplementary Tables 1 and 2). Metformin has been shown to

reduce osteoclastogenesis *in vitro*[35] but it is not clear whether it also reduces osteoclastogenesis *in vivo*, particularly in T2D. We confirmed that metformin significantly reduced the number of OCs (Fig. 5c,d) and bone resorption activity *in vitro* (Fig. 5e,f) in OC cultures derived from the bone marrow of metformin-treated MKR mice. Importantly, metformin reduced the OC surface per

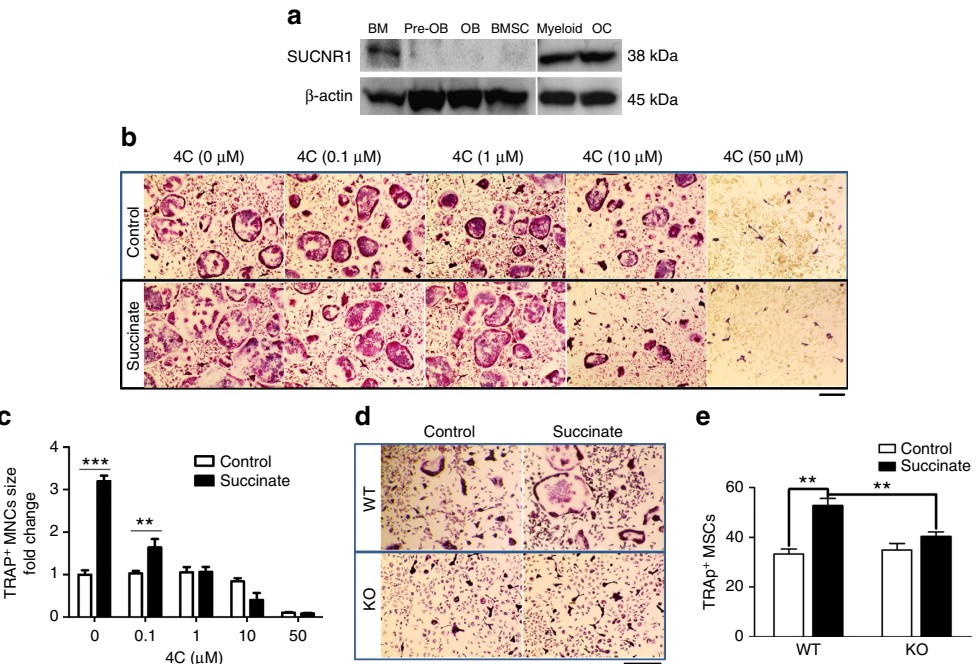

**Figure 4 | Succinate stimulates osteoclastogenesis via SUCNR1.** As described in the method, ficoll-processed non-adherent bone marrow cells were seeded at 20,000 per well in 96-well plate and stimulated with the indicated concentration of cytokines (M-CSF, 30 ng ml$^{-1}$ from day 1; RANKL, 50 ng ml$^{-1}$ from day 3); fresh cytokines were added every other day. (**a**) Western blotting of SUCNR1 in various cell lineages from bone. Representative TRAP staining images and analysis of (**b,c**) WT mice bone marrow *in vitro* OC cultures treated with 1 mM succinate or control in the presence of SUCNR1 antagonist 4c at indicated concentrations. Scale bar, 200 μm. (**d,e**) Succinate effects in the OC cultures derived from bone marrow of WT and SUCNR1 KO mice, scale bar 250 μm. Data show mean ± s.e.m. of triplicates. **$P < 0.01$, ***$P < 0.001$ according to Bonferroni *post hoc* test after ANOVA, $n = 5$.

bone surface as indicated by TRAP-positive (TRAP +) cells *in vivo* (Fig. 5g,h) and its protection on bone mass led to a greater trabecular bone area in MKR mice receiving daily administration of metformin for 14 days (Fig. 5i). Further, the protective effect of metformin on the preservation of bone volume in MKR mice was confirmed by μCT analysis (Fig. 5j–p) without altering the growth of mice (Fig. 5q). Metformin started to reduce the blood glucose levels in MKR mice after 7 and 14 days (Fig. 5r), which alleviated the severity of hyperglycaemia in MKR mice.

**Succinate regulates osteoclastogenesis via NF-κB signalling.** The reduction of succinate levels by metformin may contribute to the beneficial effects of metformin in protecting bone damage from diabetes. We investigated whether adding succinate back could rescue osteoclastogenesis in the OC cells derived from metformin-treated mice. Indeed, OC culture derived from MKR mice treated with metformin for 2 weeks yielded 20% fewer multinucleated TRAP-positive OC cells; adding succinate *in vitro* was able to restore the numbers of OCs to the same level as that of the control culture (Fig. 6a,b). Interestingly, *in vivo* metformin exhibited a minimal effect on the osteoclastogenesis derived from WT mice, in which low succinate levels were observed. This result supports the idea that metformin inhibition on osteoclastogenesis may act via antagonizing hyperglycaemia-induced succinate elevation. The classical NF-κB signalling activation mediated by nuclear localization of p65-p50 heterodimer nuclear localization plays an important role in RANKL-induced osteoclastogenesis. Succinate administration also upregulated the levels of p65 and p50 in the nuclear protein lysate from OC cells (Fig. 6c) and SUCNR1 is required for succinate stimulation of NF-κB signalling. In contrast to OCs derived from WT mouse bone marrow, succinate failed to stimulate the nuclear localization of p65 and p50 in OCs derived from SUCNR1 KO mouse bone marrow (Fig. 6d and Supplementary Fig. 6a). Similarly,

succinate-enhanced RANKL-induced NF-κB increase in RAW 264.7 cells (Supplementary Fig. 6b). Metformin moderately reduced p50 levels compared with control cells but significantly reduced the stimulation of both p50 and p65 by succinate (Fig. 6c). Interestingly, the expression of a key alternative NF-κB pathway protein RelB remained stable irrespective of succinate or metformin treatment (Supplementary Fig. 6c). These findings indicate that succinate and metformin converge on the classical NF-κB signalling pathway by antagonizing each other in regulating osteoclastogenesis.

As summarized in Fig. 7, our results demonstrate a molecular route for a metabolite-enhanced bone degradation in diabetes. The bone from T2D model with hyperglycaemia consists of succinate accumulation due to SDH deficiency in BMSCs. Succinate promotes OC differentiation and bone resorption via the activation of SUCNR1 and NF-κB signalling. Suppression of succinate action—either reducing the ligand level by metformin or targeting the receptor by SUCNR1 antagonist 4c, knocking out or knocking down with siRNA—effectively suppresses succinate-enhanced osteoclastogenesis and bone loss in hyperglycaemic conditions.

**Discussion**
The detrimental effects of hyperglycaemia have been well documented with increased oxidative stress, which also impairs bone marrow cells[36–40]. Oxidative stress results in an imbalance between reactive oxygen species (ROS) production and the capacity of the antioxidant defence system. In hyperglycaemic conditions, excess ROS produced by the inflammatory and oxidative stress microenvironment in the bone marrow may induce cell damage and the apoptosis of stem/progenitor cells and alter the bone marrow niche, resulting in impaired bone metabolism in diabetic patients[41]. The activities of oxidative enzymes in mitochondria—including SDH, malate

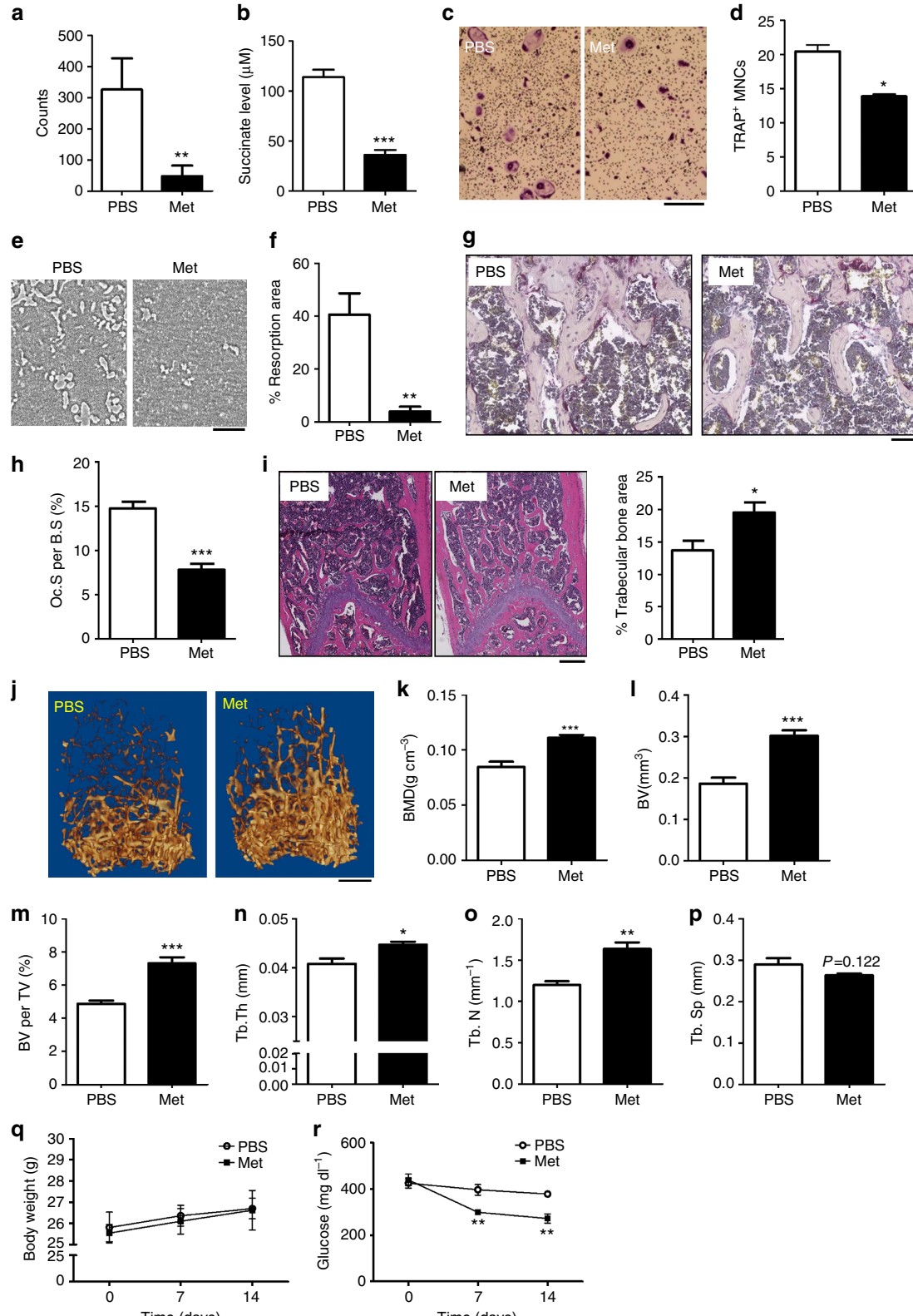

**Figure 5 | Metformin reduces the succinate level and impedes osteoclastogenesis in hyperglycaemia.** Three-month-old MKR mice were treated daily with PBS ($n = 5$) or metformin (Met, $n = 6$) (200 mg kg$^{-1}$) for 14 days. Succinate level in **a** BMSCs measured by LC-MS and (**b**) serum measured by succinic acid colorimetric assay. (**c,d**) TRAP staining (scale bar, 400 µm) and (**e,f**) osteo surface resorption assay in OC cultures on day 6 (scale bar, 100 µm). Data show mean ± s.e.m. of triplicated wells. *$P < 0.05$, **$P < 0.01$ according to a two-tailed $t$-test, $n = 3$. (**g,h**) Representative TRAP staining images (scale bar, 100 µm) and quantitative result of TRAP+ cell surface per bone surface (Oc. S per B.S) of femur proximal metaphyseal regions. (**i**) Representative H&E staining images and analysis of trabecular bone area (scale bar, 250 µm). (**j**) Representative µCT three-dimensional structures (scale bar, 500 µm) and (**k–p**) BMD, BV per TV, Tb.N, Tb.Th, and Tb.Sp analysis. (**q**) Body weight and (**r**) glucose levels over 14 days. Data shown mean ± s.e.m. *$P < 0.05$, **$P < 0.005$, ***$P < 0.001$ according to a two-tailed $t$-test. BMD, bone mineral density.

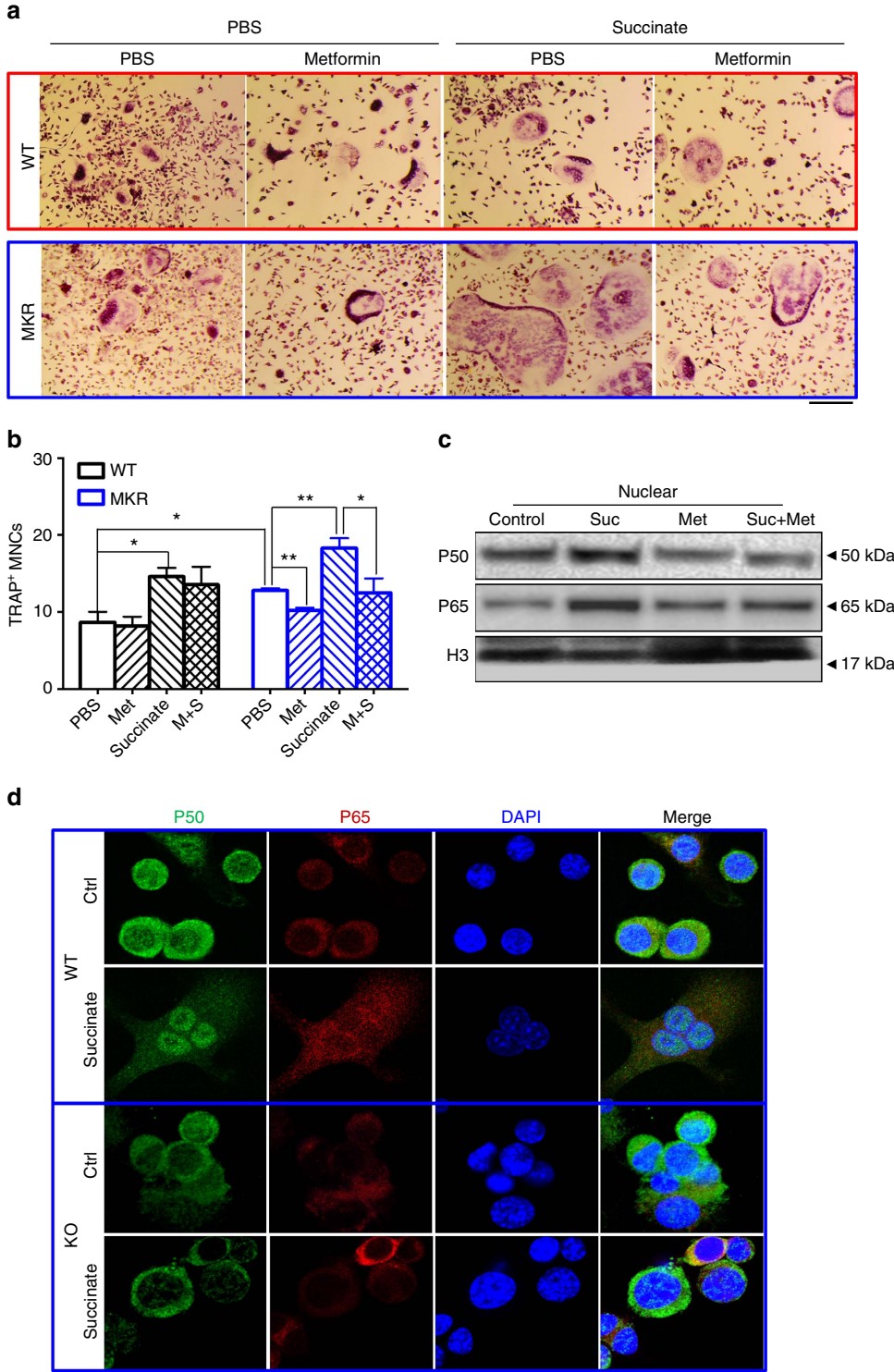

**Figure 6 | Succinate-enhanced osteoclastogenesis via NF-κB activation is counteracted by metformin.** (**a**) Representative TRAP staining images (scale bar, 250 μm) and (**b**) analysis of OC cell culture on day 6 derived from WT and MKR mouse bone marrow cells. PBS or metformin (200 mg kg$^{-1}$) were administrated daily to the mice for 14 days before bone marrow cells were harvested for OC cultures; succinate: 1 mM succinate was added in OC cell cultures; data show mean ± s.e.m. of triplicated wells; *$P < 0.05$, **$P < 0.01$ according to a two-tailed $t$-test post ANOVA, $n = 4$. (**c**) Western blotting of OCs culture derived 12-week-old MKR mice. Suc: 1 mM, Met: 500 μM. Levels of p50 and p65 of nuclear proteins from bone marrow OC cells. (**d**) Indirect IF of P65, P50 and DAPI (4′,6-diamidino-2-phenylindole) in OCs derived from WT and SUCNR1 KO mouse bone marrow cells stimulated with 30 ng ml$^{-1}$ MCSF and 30 ng ml$^{-1}$ RANKL on day 5.

dehydrogenase, glutamate dehydrogenase, and isocitrate dehydrogenase—are closely related to ROS production[42–45]. The accumulation of succinate, malate, glutamate and citrate that is observed in T2D BMSCs (Supplementary Table 1) indicates that hyperglycaemia impairs the activities of these enzymes in bone. Reduced SDH activity and succinate

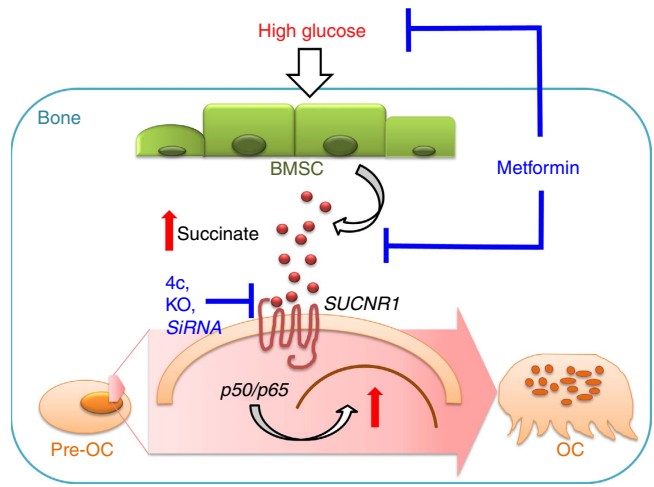

**Figure 7 | Schematic diagram summarizing the mechanisms of succinate induced osteoclastogenesis with hyperglycaemia.** Under hyperglycaemic condition, BMSCs accumulate and release succinate into the bone microenvironment. Succinate binds to its G-protein coupled receptor SUCNR1 to stimulate OC differentiation and bone resorption. Targeting succinate action either through the activation or expression of SUCNR1 or reduction of succinate level, effectively blocks osteoclastogenesis activated by succinate in hyperglycaemic conditions.

accumulation were also observed in the kidneys of diabetic rats[46]. In addition to SDH deficiency, the non-enzymatic conversion of α-ketoglutarate to succinate under oxidative stress may also contribute to succinate accumulation[47]. However, such an ROS auto-regulatory feedback loop would cause a buildup of α-ketoglutarate[48] that was not detected in MKR mice (Supplementary Table 1). We conclude that the accumulation of succinate in BMSCs should be largely due to SDH deficiency with hyperglycaemia (Fig. 1e,f).

Changes in mitochondrial enzymes are fundamental to osteogenic activity because these enzymes are part of the aerobic hydrogen transport system from which rich-energy bond phosphate is generated[49]. However, the mechanism underlying the actions of the accumulated metabolites in bone remodelling remains elusive. Succinate as an extracellular signal molecule in regulating bone remodelling sheds light on the connection between cellular metabolites and organ metabolism. Succinate has been demonstrated as a class of metabolites with an inflammatory signalling capacity[50]. We observed that the nuclear levels of p65 and p50, increased in OCs after succinate stimulation (Fig. 6). Recently, HIF-1α, which is associated with NF-κB activation[51], has been found to play a key role in glucose metabolism and OC bone resorption[52]. Interestingly, succinate was shown to directly inhibit prolyl hydroxylase domain enzyme activity in macrophages, resulting in the stabilization of HIF-1α and the induction of a range of target genes[53]. In addition, patients harbouring mutations in SDH have increased levels in both HIF-1 α activity and circulating succinate[53,54]. Notably, the sustained succinate increase also induces the pro-inflammatory cytokine IL-1β via HIF-1α (refs 50,53). Indeed, we observed the elevation of IL-1β in succinate-treated mice (Fig. 3l and Supplementary Fig. 4i). These data suggest that succinate may directly regulate HIF-1α signalling and pro-inflammatory cytokines to stimulate osteoclastogenesis. The regulation of succinate with HIF-1α and pro- inflammatory cytokines such as TNF (Fig. 3m and Supplementary Fig. 4h) warrants additional studies to further reveal the action of succinate and its interactions with other known osteoclastogenic regulators during osteoclastogenesis and bone resorption.

Since the characterization of SUCNR1 as the receptor for succinate[18], it has been demonstrated to modulate a vast array of functions in the human liver, spleen, intestines and kidneys. Here, we reveal a function of succinate in bone as a stimulator of osteoclastogenesis via SUCNR1 signalling both in vitro (Fig. 2) and in vivo (Fig. 3 and Supplementary Fig. 4). Our data suggest that succinate may enhance bone loss through its stimulation of osteoclastogenesis without increasing the pool of the OC precursor cells (Fig. 3h and Supplementary Fig. 5c). SUCNR1 seems essential for succinate stimulation of osteoclastogenesis since the stimulation was blunted by SUCNR1 antagonist and siRNA and abolished in OCs derived from SUCNR1 KO mouse bone marrow (Fig. 4). Furthermore, we demonstrated that targeting succinate actions by metformin, which reduced succinate level (Fig. 1 and Supplementary Table 2), effectively blunted osteoclastogenesis (Fig. 5).

In addition to diabetes itself as a risk factor for osteoporosis, the drugs used to treat T2D often adversely impact bone. Thiazolidinedions clearly decrease bone mass via their actions on the peroxisome proliferation-activated receptor gamma (PPARγ)[55]. Activation of PPARγ inhibits the differentiation of osteoblasts and enhances the differentiation of OCs and adipocytes. Two members from the most recently approved class of anti-diabetic drugs, inhibitors of sodium-glucose co-transporter 2 (SGLT2), have also been associated with fracture increases. SGLT2 inhibitors decrease the concentrations of plasma glucose by inhibiting proximal tubular resorption of glucose in the kidneys. Their potential mechanism of adverse effects in bone may involve increased serum phosphate concentrations and parathyroid hormone concentration with SGLT2 inhibitors[56]. Notably, metformin, a widely used biguanide medicine for treating T2D, is the only anti-diabetic drug that has been reported to have neutral or favourable effects in terms of bone preservation. In T2D patients, metformin treatment has been associated with a decreased risk of bone fracture[57]. In osteoblast cultures obtained from rat calvaria, metformin increased trabecular bone nodule formation[58]. In ovariectomized rats, metformin has been shown to improve compromised bone mass and quality[59]. In an STZ-induced T1D model, metformin stimulated bone lesion regeneration in rats[60]. Metformin inhibits OC differentiation and reduces the number of OCs by reducing RANKL expression[35,61] and inhibition of NF-κB activation[62]. Our study sheds light on a new mechanism to understand how metformin treatment influences skeletal tissues in the type 2 diabetic condition from a metabolic perspective.

Interestingly, 3-nitropropionic acid (3-NPA), a molecule that inhibits SDH activity[63] and leads to succinate accumulation intracellularly, impedes osteoclastogenesis (Supplementary Fig. 2d). The role of succinate as a metabolite intracellularly differs from its role as an extracellular signal molecule. It is possible that other metabolites also have functions beyond their role as intermediate metabolites. BMSCs from MKR mice exhibit a 2.5-fold increase in glutamate levels (Supplementary Table 1). Diabetes patients have increased glutamate levels[64], and it has been shown that high extracellular glutamate inhibits osteoblastic cell proliferation[65], suppresses osteoblastogenesis from mesenchymal stem cells[66], and stimulates osteoclastogenesis[67].

Our study may lead to a better understanding of the fundamental mechanisms of complications in metabolic diseases. It is anticipated that the potential role of succinate and other metabolites in the pathogenesis of hyperglycaemia or other metabolic diseases-induced bone defects will be assessed. Moreover, since we demonstrated the efficacy of blocking succinate and SUCNR1 signalling in preventing osteoclastogenesis, metabolites may represent a new angle for therapeutic targets

preventing bone damages associated with succinate elevation in diabetes, and other conditions involving SDH activity deficiencies such as cancer and aging.

## Methods

**Reagents.** Metformin was from Calbiochem (Darmstadt, Germany). Dulbecco's Phosphate-Buffered Saline, Succinate (Succinic acid) and Ponceau S solution were from Sigma-Aldrich (St Louis, MO, USA).The Succinate Colorimetric Assay Kit (catalogue number: K649) was from BioVision (Milpitas, CA, USA). Ten per cent buffered formalin was from Fisher Scientific (Hampton, NH, USA). Recombinant murine sRANK ligand (RANKL) and macrophage colony-stimulating factor (M-CSF) were from PeproTech (Rocky Hill, NJ, USA). Antibodies against p50, p65, β actin, DAPI (4′,6-diamidino-2-phenylindole) and Histone H3 were purchased from Cell Signalling (Danvers, MA, USA), SUCNR1 was from Novus Biologicals (Littleton, CO, USA). The SUCNR1 antagonist 4c (4c) was synthesized by Shenyang Wuhe BioTech Co. (Liaoning, China) based on the previous study[34]. Details on the (monoclonal) antibody target, clone name, source and dilution used for all antibody based methods can be found in the Supplementary Table 3.

**Animals.** All animal experiments related to this study have been approved by the Institutional Animal Care and Use Committee (IACUC) under protocol number: 160108 and 170405, and performed in accordance with Division of Laboratory Animal Resources (DLAR). All the animals were housed in Specific Pathogen-Free units at the New York University animal facility. Homozygous MKR transgenic mice were determined by PCR by genotype and confirmed by high glucose phenotype ($>350\,mg\,dl^{-1}$ at 8-week-old) in the breeders and offspring. The same background FVB/NJ WT mice (catalogue number: 001800) used in this study were purchased from The Jackson Laboratory (Bar Harbor, ME, USA). Age-matched male WT and MKR mice were randomly assigned to two groups ($n=5$ in each group). Vehicle (PBS) or metformin (Met, $200\,mg\,kg^{-1}$ BW) in a 50-µl volume were introduced each day via intraperitoneal (I.P.) injections for 14 days. The succinate was neutralized with sodium hydroxide to achieve a solution with a pH of 7.4 and I.P. injected at $4\,mM\,kg^{-1}$ per day in a volume of 100 µl. C57B/6 strain WT and Sucnr1 KO mice were derived from $Sucnr1^{+/-}$ breeders originally provided by Novartis.

**Bone marrow stromal cell culture.** The bone marrow cells flushed out from long bones were cultured in MEM Alpha Modification (α-MEM) medium containing L-Glutamine, Ribo-, and Deoxyribonucleosides (HyClone, Logan, UT, USA), supplemented with 15% Fetal Bovine Serum (Atlanta Biologicals, Flowery Branch, GA, USA), $100\,\mu g\,ml^{-1}$ streptomycin and 100 units $ml^{-1}$ penicillin (Gibco, Grand Island, NY, USA) in a 37 °C, 5% (v/v) $CO_2$ humidified incubator. After non-adherent cell removal, the adherent cells were cultured as BMSCs for 7 days and harvested into Eppendorf Protein LoBind tubes for metabolomics assay.

**Metabolomics sample preparation.** At the time of extraction for HPLC, BMSCs were mixed with cold 80% methanol (mass-spec grade) (Fisher Scientific) at a ratio of 30 µl per million cells, and then quickly thawed on heat-block set at 50 °C for 5 min. The suspension was then processed with three rounds of 1 min vortexing at max speed and then chilled briefly on dry ice. The mixture was incubated at 4 °C for 1 h before centrifuge at $20,000g$ for 20 min at 4 °C. The supernatant was stored at $-20$ °C and used as metabolite extract for the LC-MS analysis. For the LC-MS analysis, the metabolite extract was transferred to a 150-µl deactivated glass insert housed in Waters 2-ml brown MS vials (Waters Corporation, Milford, MA, USA). A chemical standard solution was prepared from synthetic complete mixture from Sigma-Aldrich (Y1501) at a concentration of $19\,\mu g\,ml^{-1}$ 80% methanol (mass-spec grade).

**LC-MS acquisition.** We analysed the metabolite extract in a platform that consisted of a Waters UPLC-coupled Exactive Orbitrap Mass Spectrometer (Thermo Scientific, Waltham, MA, USA), using a mix-mode OPD2 HP-4B column ($4.6 \times 50\,mm$) (Shodex, Showa Denko, Tokyo, Japan). The column temperature was maintained at 45 °C. Five microlitres of each sample maintained at 4 °C were loaded by the autosampler (Fisher Scientific) in partial loop mode for three times at positive mode and negative mode, respectively. The binary mobile phase solvents were: A, 10 mM NH4OAc in 10:90 Acetonitrile:water; B, 10 mM NH4OAc in 90:10 Acetonitrile:water. Both solvents were modified with 10 mM HOAc for positive mode acquisition, or 10 mM NH4OH for negative mode. The 30 min gradient for both modes was set as: flow rate, $0.1\,ml\,min^{-1}$; 0–15 min, 99% A, 15–18 min, 99% to 1% A; 18–24 min, 1% A; 24–25 min, 1% to 99% A; 25–30 min, 99% A. The MS acquisition was in profile mode and performed with an ESI probe, operating with capillary temperature at 275 °C, sheath gas at 40 units, spray voltage at 3.5 kV for positive mode and 3.1 kV for negative mode, capillary voltage at 30 V, tube lens voltage at 120 V, and Skimmer voltage at 20 V. The mass scanning used 100,000 mass resolution, high dynamic range for the AGC Target, 500 milliseconds as the Maximum Inject Time, and $75-1,200/mz$ as the scan range. The system was operated using Thermo Xcalibur v2.1 software (Thermo Scientific). All of the chemicals were from Sigma-Aldrich unless otherwise specified.

**Succinate colorimetric assay.** Blood was collected via cardiac puncture after the animals were euthanized, and the blood was left at room temperature for at least 30 min before centrifuging at $200g$ for 10 min to allow the sera to separate. The bone marrow was flushed with PBS; after being washed twice, the total bone marrow was re-suspended in 500 µl of PBS. We used a Succinate (Succinic Acid) Assay kit (BioVision Inc.) to detect the succinate levels in serum, bone marrow flush, and BMSC samples. 50 µl samples were added into duplicate wells of a 96-well plate and the assay was performed following the manufacturer's protocol.

**Osteoclastogenesis and resorption assays.** The bone marrow was flushed from tibia and femur of 8–16-week-old mice with serum-free α-minimum Eagle's medium, and the marrow clumps were broken down by passing the media through a syringe with a 19-gauge needle. After centrifugation, the cells were purified over the Ficoll-Paque (Ficoll-processed) or cultured directly. The cell pellets were re-suspended in α-minimum Eagle's medium containing 10% FBS and incubated at 37 °C in 5% $CO_2$. The cells were seeded at 20,000 per well in 96-well plate and stimulated with the indicated concentration of cytokines (M-CSF, $30\,ng\,ml^{-1}$; RANKL, $50\,ng\,ml^{-1}$) from the time of culture unless otherwise noted in the figure legends; fresh cytokines were added every other day. To perform the TRAP staining, the cell cultures were fixed for 5 min and then stained with Naphthol AS-BI phosphate and a tartrate solution for 15 min at 37 °C, followed by counterstaining with a hematoxylin solution. Multinuclear ($>3$ nuclei) TRAP + cells were identified and counted as OCs using light microscopy. For the resorption assay, the cells were plated into Osteo Assay Surface plates (Corning, NY, USA), which mimic *in vivo* bone surface, and the same culture procedure was used as previously described for the osteoclastogenesis assay. We used 10% bleach to wash away the cells after OC culture, and we calculated the pit formation ratio (pit area versus total area) based on the images captured using an EVOS cell imaging system (Life Technologies, Carlsbad, CA, USA).

**µCT and bone histomorphometry.** We evaluated the bone samples by µCT using a SkyScan 1172 high-resolution scanner (Brucker, Billerica, MA, USA) with a 60 kV voltage and a 167 µA current at 9.7 µm resolution. The femur metaphyseal region between 0.242 mm (25 slides) and 2.745 mm (283 slides) from the growth plate was selected as regions of interest (ROIs) for trabecular bone measurement. Binarized ROIs from each animal were then used to calculate the bone mineral density followed by three-dimensionsal analysis using CTAn (Brucker) to collect morphometric parameters: bone volume (BV), relative bone volume (BVper TV), trabecular number (Tb.N), trabecular thickness (Tb.Th), and trabecular separation (Tb. Sp)[68]. CTVox (Brucker) was used to generate the three-dimensional images of the ROIs. For the histomorphometry and peripheral quantitative computed tomography, the tibiae and femurs were preserved in 70% ethanol until they were decalcified with 10% EDTA for paraffin embedding. We performed hematoxylin and eosin (H&E) and TRAP staining using an Acid Phosphatase Kit 387-A (Sigma-Aldrich). We analysed the bone histomorphometry parameters using open-source software[69] and image J.

**Flow cytometry analysis.** Mouse BM was flushed with PBS from long bones. After hypotonic red cell lysis, cells were blocked with 2.4 G2 antibody followed by incubation with a mix of Ly6C-PE-Cy7 (clone AL-21; BD Biosciences Pharmingen), and CD11b-APC (clone M1/70; BD Biosciences Pharmingen). Flow cytometry was performed on a FACSCalibur (BD Biosciences) and analysed with FlowJo software (Tree Star Inc.). Cells were gated based on CD11b and Ly6C staining intensity.

**Serum cytokine and resorption marker measurement.** We determined serum IL-4 and IL-13 levels using the Mouse Cytokine Antibody Array (R&D Systems, Inc., Minneapolis, MN, USA) in duplicate; we quantified the density using Image J software and normalized it to that of the positive control. We measured serum resorption marker TRAP5b (TRAP) with MouseTRAP (TRAcP 5b) ELISA (Catalogue number: SB-TR103, Immuno Diagnosticsystems, Gaithersburg, MD). TNF and IL-1β levels were determined with Quantikine ELISA Kits from R&D Systems (Catalogue number: MTA00B and MLB00C) following the instructions from the manufacturers.

**Western blotting.** We extracted the total protein using the RIPA lysis buffer (Thermo Scientific), and we isolated the nuclear protein using an EpiQuik Nuclear Extraction Kit (Catalogue number: op-0022-100, Epigentek, Farmingdale, NY, USA) and used a Pierce BCA Protein Assay Kit (Catalogue number: 23,225, Thermo Scientific) to determine the protein concentrations. Protein samples were denatured in SDS sample buffer (2% SDS, 62.5 mM Tris-base (pH 6.8), 10% glycerol, 5% β-mercaptoethanol, and 0.005% bromophenol blue) and loaded into a 10% SDS–polyacrylamide gel electrophoresis gel (Fisher Scientific). The gel was preceded according to a standard Polyvinylidene difluoride membrane

transfer, incubated with primary and secondary antibodies, and visualized using peroxidase substrate for enhanced chemiluminescence (Thermo Scientific) on ChemiDoc XRS System (BioRad Laboratories, Inc. Hercules, CA, USA). All the original digital images can be found in the Supplementary Figs 7–9.

**Indirect immunofluorescence.** Ficoll-processed WT and SUCNR1 KO mouse bone marrow cells were seeded at $1.5 \times 10^5 \, ml^{-1}$ cells per chamber (eight-chamber per slide) with $20 \, ng \, ml^{-1}$ MCSF added from day 1 and $30 \, ng \, ml^{-1}$ RANKL added from day 3. Primary antibodies of P65, P50 and DAPI and fluorophore-conjugated secondary antibody against the primary antibody were used for detection of protein localization on day 5 after fixation. The images were captured by Zeiss LSM880 confocal microscope, using a Zeiss Plan-Apochromat 63x/1.40 Oil lens.

**Real-time PCR.** We extracted RNA from the OC cells using an RNeasy Plus Mini Kit (Catalogue number: 74136, Qiagen, Venlo, Netherlands). RNA (500 ng) was converted into cDNA using a Reverse Transcriptase Kit (Catalogue number: N8080234, Fisher Scientific) according to the manufacturer's instructions. The primers that we used included: mNFATc1 Fr: 5′-CCGTCACATTCTGGTC CATAC-3′; mNFATc1 Rv: 5′-CCAATGAACAGCTGTAGCGTG-3′; mTRAP Fr: 5′-TGTGAGGGAGGAGGCGTCTGC-3′; mTRAP Rv: 5′-CGTTCCCAAGAAA GCTCTACC-3′; mCtsk Fr: 5′-GTGTCCATCGATGCAAGCTTGGCA-3′; mCtsk Rv: 5′-GCTCTCTCCCCAGCTGTTTTTAAT-3′; CalR Fr: 5′-GCTCTGCGAGGG GATCTATCT-3′; CalR Rv: 5′-CTGCACTCAGCCAGCAGTTGT-3′; mFTH1-Fr: 5′-CACCATGACCACCGCGTC-3′; mFTH1-Rv: 5′-AGACGTAGGAGGCATACA ACT-3′. We used a CFX384 Touch Real-Time PCR Detection System (Bio-Rad, Philadelphia, PA, USA).

**Statistical analysis.** We used analysis of variance when the study subjects included more than two groups, followed by the Bonferroni $t$-test. We used the two-tailed Student's $t$-test to compare the difference between two experimental groups. A value of $P < 0.05$ was considered to be statistically significant.

**Data availability.** Metabolomics data that support the findings of this study have been deposited in Metabolomics Workbench with the identifier 'http://dx.doi.org/10.21228/M8MG6K'. The authors declare that all other data supporting the findings of this study are available in the article and its Supplementary Information files or from the authors upon request.

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

## Acknowledgements

We thank Dr Derek LeRoith for providing the MKR mice and Drs Laurie McCauley and Nicola Partridge for their constructive criticisms. This work was supported by National Institutes of Health (NIH) grants R01CA180277 and R03 CA172894 to Xin Li, 5R01GM06248012-12, 3U54DK10255602S2 grant and California Institute for Regenerative Medicine grant (RB4-06087) to Michael Snyder, grants from Ministry of Science and Technology of China (No.2014DFA32120), the National Natural Science Foundation of China (No.81471000), and the Natural Science Foundation of Liaoning (No.2014023042) to Yingjie Wu, and NYU Provost Office Mega Grant Seed Fund Initiative to Deepak Saxena and Xin Li.

## Author contributions

Y.G. performed the majority of the experiments, analysed the data, maintained the animals and collected the samples. C.X. performed the antagonist experiments and analysis. Xiyan L. performed the metabolomics experiments and analysis. T.Y. assisted with the *in vivo* experiments. J.Y. performed the SUCNR1 western blotting. T.Z. helped with the *in vitro* bone resorption assay and bone histologic analyses. M.S. and Y.W. provided reagents and suggestions for the project. C.X., Xiyan L. and D.S. helped with the manuscript preparation. Xin L. supervised the project and wrote the manuscript.

## Additional information

**Competing interests:** The authors declare no competing financial interests.

