## [Peer Review File · Nature Communications]

Reviewers' comments:

Reviewer #1 (Remarks to the Author):

This paper identifies succinate as a mediator of osteoclastogenesis in Type 2 diabetes. Succinate is shown to be elevated in models (including in human bone marrow stromal cells treated with high glucose and acting via its receptor SUCRN1) it can promote bone loss in vitro and in vivo. The authors also demonstrate that its effects are countered by metformin. These are interesting data. There are however some issues which need to be addressed to support the conclusions being drawn.

1. In Figure 1D we see high glucose driving an increase in succinate. This is of interest but what might the mechanism be? Are other TCA cycle intermediates similarly increased or is there a particular increase in succinate? The authors should go at least some way to addressing mechanism here as it is critical for their overall conclusions regarding hyperglycemia and bone loss.
2. In Fig 3 succinate is shown to down-regulate IL4 and IL13 in serum. What about other cytokines such as RANKL, IL-1beta, TNF or IL6? We might expect the osteoclastogenic cytokines to be elevated.
3. Figure 4 is a critical figure in the paper in my opinion, as it implicates SUCRN1 in the effect of succinate. The only evidence we have is the use of a SUCRN1 antagonist 4c. I notice that the authors have published on this compound previously but even still we need more information here. Can the authors test 4c in the assays shown against another inducer of TRAP staining to ensure specificity? Can they demonstrate antagonism of succinate binding to the cells used here? Also I would like to see these data supported by another approach - eg the use of a blocking antibody to SUCRN1. Finally since SUCRN1-deficient mice are available and indeed mentioned in the discussion, they should be tested in the in vivo model. These experiments will give us confidence that the effect of succinate here is indeed via SUCRN1.

Reviewer #2 (Remarks to the Author):

This is a well written manuscript examining the role of metabolic changes, specifically succinate, in type 2 diabetic osteoclastogenesis. The influence of metabolic changes on bone remodeling is an important area of investigation. Here the authors identify an increase in succinate levels in MKR mice as well as in bone marrow cultured under high glucose conditions. The MKR mice displayed an increase in osteoclasts. Further studies show in vitro that succinate treatment increases osteoclastogenesis and decreases bone density in mice. An inhibitor to the succinate receptor shows a trend to decrease succinate induced osteoclastogenesis in vitro. Metformin is also used to show a decrease in succinate in the MKR mice which display an increase in bone density. In addition, bone marrow cells obtained from mice treated with metformin show decreased osteoclastogenesis when treated with succinate in vitro. The studies provide a variety of approaches to address the role of succinate in osteoclastogenesis. However, the link isn't fully demonstrated at this point and further studies are needed. In particular, the studies would greatly benefit from additional cohort studies, blood glucose/body weight and bone density data, additional in vivo osteoclast measures and optimally a study showing that blocking succinate receptor in vivo prevents bone loss resulting from diabetes or succinate treatment.

Major comments:

- 1) Type 2 diabetes is not usually associated with bone loss. If type 2 diabetes is associated with elevated succinate levels which increase osteoclastogenesis, then one wonders how trabecular bone density is maintained. More information/discussion on how this could occur would be helpful in the introduction; this will set a stronger basis for the studies.

- 2) General information on the mice in each experiment is critical for interpreting the results, but not included. It is important to show the mouse blood glucose or HbA1C levels as an indicator of diabetes and its severity – this is never shown in any of the studies and is an important control. Similarly, mouse weights should be shown for each study since weight can influence bone density. In addition, because there are several mouse experiments, it would be helpful to have the mouse ages in the figure legends.
- 3) Information on when succinate increases during diabetes development in the MKR mice would be helpful, if possible. Does it increase immediately with diabetes onset?
- 4) Bone density should be included in the first study (Figure 1), since the authors are showing increased osteoclastogenesis. Does this lead to bone loss?
- 5) In figure 1 the only measure of activated osteoclasts is histological, additional measures such as serum TRAP would strengthen this data. It is also critical to show that the results reproduce in another cohort of mice; it would be best to increase the n in the second run since an n of 4 seems quite low.
- 6) Figure 1 legend is wrong – 1C legend discusses BMSCs from WT and MKR mice, but they show serum succinate levels.
- 7) Figure 2B: When doing t-test, what is being compared? Are all conditions compared? When counting TRAP pos cells, is the final number per well? Per view? Per mm²?
- 8) Figure 2 – given that the serum contains around 100um of succinate, it would be useful to quantitate the osteoclast numbers in the succinate dose response which ranges from 100um (in diabetic serum) to 1mM (which is used in all in vitro studies).
- 9) Figure 3 – Is the succinate solution endotoxin free? Succinate may increase inflammation leading to osteoclast activation. It would be helpful to use the antagonist or a receptor KO to prove that succinate is directly working through the receptor.
- 10) Figure 3 – similar to past comments, the data would be strengthened by additional measures of osteoclast activity (ie, serum markers) as well as including an additional cohort of animals that repeats the findings.
- 11) Figure 3 – Were other cytokines measured? What were their levels? Can the measured cytokines be given a concentration? It is unclear what the “density” means.
- 12) Figure 4 – The receptor antagonist did not significantly decrease TRAP MNCs (#/???) in figure 4C. Also, in figure 4E succinate did not significantly increase TRAP MNCs, suggesting variability between repeat studies. Statistics need to use a post-hoc anova.
- 13) It would be helpful if a rationale was given for the treatment periods, ie: why 14 days treatment? Were other shorter time points tried that did not have an effect?
- 14) Figure 5 – Again, there aren't any mouse parameters such as blood glucose, body weight. Metformin could affect hyperglycemia and the succinate changes may be secondary to this.
- 15) What about using WT mice that are treated with succinate and then giving these mice metformin? This would remove the hyperglycemia component.
- 16) Figure 6 – Blood glucose levels of treated mice?? This study was a bit confusing since the mice were treated with metformin and then bone marrow stromal cells isolated and then treated with succinate. Figure 6C should be noted as WT OC cultures? It isn't fully clear what is going on in this figure.

Reviewer #3 (Remarks to the Author):

The paper reports that succinate enhances osteoclast formation, most likely through activation of the previously orphan GPCR, now known to be the succinate receptor (SUCNR1). They used MKR mice which rapidly develop hyperglycaemia due to type II diabetes. They recently published data obtained with the use of metabolomics, showing in bone marrow, elevated levels of TCA cycle metabolites, particularly succinate, which is the object of this study.

Part of this paper confirms the recent publication (reference 13), showing the great increases in

succinate produced by BMSCs from hyperglycaemic animals. The present paper also they show greatly elevated plasma levels of succinate to the order of 100 μ M, as well as decreased bone with increase in osteoclast parameters.

In view of the known increase in osteoclasts and resorption in diabetic bone disease they investigated the possible role of succinate. There were some known findings that seemed to make this worthwhile. For example, succinate through SUCNR1 was known to promote NF κ B translocation, to increase intracellular Ca in an IP-dependent manner via PLC β , and to inhibit PGE-increased cAMP (ref 15).

The rationale of the work is clearly explained, but it suffers much from poorly explained methods and experiments, among other problems.

Specific matters.

1. Figure 2, several questions.

In the text and in several figure legends, the term "primary OC cultures" is used. This is wrong and misleads. These are primary BM cultures treated with RANKL and M-CSF to generate osteoclasts over several days, and that should be made clear in every instance.

Indeed the description of relevant method is deficient. BM cells are removed and plated out etc, but there is no statement of the size of culture wells or the cell density. This is particularly important in these experiments because BM cultures of a certain density will have sufficient accompanying stromal cells to mediate differentiation of osteoclasts through RANKL production with appropriate treatment. Here it is just not clear. They appear to treat with RANKL and M-CSF in all experiments, but it is not clear whether they need it in all – they never state what the cell density etc is, and particularly, as I refer to below, in all their experiments the actual number of osteoclast generated appears very low – what were the size of the wells in the density of cultures etc? The data it is therefore extremely difficult to interpret without further details.

2. Figure 3. Were these bone data obtained with $n = 5$ per group? This is a small number for such experiments particularly when such small effects were obtained. Were the animals pair-fed? Any effect of succinate on feeding and hence on growth over the seven weeks?

Fig 3K. I am assuming that this is the result of OC measures in bone sections. If it is ex vivo generation from narrow, exact details would be required. If it is in vivo data it is given as "TRAP+ve/mm". More important is a measure of OC surface.

Why select IL-4 and IL-17 from among the many cytokines that either inhibit or increase osteoclast formation? This is not interpretable, quite apart from the fact that the IL-4 data is not significant.

3. 4B, C. 1mM succinate was used to treat the primary marrow cultures (? Also treated with RANKL, M-CSF?? Not stated). Again, the number of OCs generated is very small indeed. Importantly though, if 10 μ M of a SUCNR1 antagonist is considered to inhibit specifically a 1mM concentration of agonist, this would require some pharmacological explanation, as it would be quite remarkable.

4D. In this experiment they are comparing the numbers of osteoclasts generated by ex vivo BM cultures from MKR mice with or without SUCNR1 antagonist treatment in vivo. Again there are very few OCs generated and a small effect claimed. Comparison of the responses in the control versus MKR requires a 2-tailed ANOVA to compare responses, and not just a comparison of the levels obtaining with the two treatments as the authors have done.

Besides, such an experiment carried out on cells in vitro needs very careful equivalence of starting numbers of BM cells in the culture. Methods do not go into this at all.

4. Fig 5 .

Metformin appeared to reduce OC formation in vitro of MKR mice cells by about 20%, but appeared to have a greater effect on resorption in vitro. This method approach is seriously questioned. According to methods, the resorption was measured at the end of a period in which

OCs were generated on the surface to be resorbed. Therefore the findings at the end of culture was actually dependent on the numbers of osteoclasts generated. The usual way such a question is addressed is to generate the osteoclasts, and transpose equal numbers of them onto bone or dentine slices and measure their ability to resorb.

Contributing to the discrepancy in this case might also be the artificial resorption substrate used, which, unlike dentine or bone can "resorb" in response to macrophages and leucocytes.

5. There is available a mouse with globally knocked out SUCNR, that is viable and healthy, and could be used to get much nearer to a definitive answer to their main research question.

We appreciate the constructive comments and believe our manuscript has been improved significantly after addressing the referees' comments. Please check the following for point-by-point response to these comments.

Reviewers' comments

Reviewer #1 (Remarks to the Author):

This paper identifies succinate as a mediator of osteoclastogenesis in Type 2 diabetes. Succinate is shown to be elevated in models (including in human bone marrow stromal cells treated with high glucose and acting via its receptor SUCRN1 it can promote bone loss in vitro and in vivo. The authors also demonstrate that its effects are countered by metformin. These are interesting data. There are however some issues which need to be addressed to support the conclusions being drawn.

1. In Figure 1D we see high glucose driving an increase in succinate. This is of interest but what might the mechanism be? Are other TCA cycle intermediates similarly increased or is there a particular increase in succinate? The authors should go at least some way to addressing mechanism here as it is critical for their overall conclusions regarding hyperglycemia and bone loss.

Response 1: We have shown that hyperglycemia reduced the activity of succinate dehydrogenase (SDH) in BMSCs (Fig. 1E-F). It is known that insufficient SDH activity leads to succinate accumulation. Some other TCA cycle intermediates are also increased but succinate is the one increased the most. It is likely that other metabolic enzymes are also affected by hyperglycemia. We will further investigate this in our future study.

2. In Fig 3 succinate is shown to down-regulate IL4 and IL13 in serum. What about other cytokines such as RANKL, IL-1beta, TNF or IL6? We might expect the osteoclastogenic cytokines to be elevated.

Response 2: We didn't detect significant changes in other cytokines by the original cytokine array which we noticed many cytokines were not detected well due to the dilution and/or low sensitivity. So we conducted new experiments using individual cytokine ELISA kits. We found osteoclastogenic cytokines such as IL-1beta and TNFalpha were significantly upregulated in serum from mice treated with succinate (Figure 3L-M and Figure S4H-I). In addition, we measured serum TRAP5b levels in a new cohort of mice and confirmed that succinate stimulated osteoclast activity (Figure S4E).

3. Figure 4 is a critical figure in the paper in my opinion, as it implicates SUCRN1 in the effect of succinate. The only evidence we have is the use of a SUCRN1 antagonist 4c. I notice that the authors have published on this compound previously but even still we need more information here. Can the authors test 4c in the assays shown against another inducer of TRAP staining to ensure specificity? Can they demonstrate antagonism of succinate binding to the cells used here? Also I would like to see these data supported by another approach - eg the use of a blocking antibody to SUCRN1. Finally since SUCRN1-deficient mice are available and indeed

mentioned in the discussion, they should be tested in the *in vivo* model. These experiments will give us confidence that the effect of succinate here is indeed via SUCRN1.

Response 3: TRAP staining is a marker for osteoclasts and RANKL is the essential stimulator of osteoclast differentiation. In the absence of RANKL, few TRAP positive osteoclasts can form *in vitro*. Therefore, RANKL has been added to OC culture in all experiments. We repeated our *in vitro* experiment with the antagonist 4c (Figure 4B-C). In addition, we took two more approaches to prove the role of SUCRN1 signaling in osteoclastogenesis. We used siRNA to knock down SUCRN1 (Figure S5D) and osteoclast culture from SUCNR1 knockout (KO) mice (Figure 4D-E). In the absence of SUCNR1, succinate was not able to stimulate osteoclastogenesis (Figure 4D-E) and the knockdown of SUCNR1 blunted the succinate stimulatory effects in osteoclast culture (Figure S4D) which strongly support the effect of succinate is via SUCRN1.

Reviewer #2 (Remarks to the Author):

This is a well written manuscript examining the role of metabolic changes, specifically succinate, in type 2 diabetic osteoclastogenesis. The influence of metabolic changes on bone remodeling is an important area of investigation. Here the authors identify an increase in succinate levels in MKR mice as well as in bone marrow cultured under high glucose conditions. The MKR mice displayed an increase in osteoclasts. Further studies show *in vitro* that succinate treatment increases osteoclastogenesis and decreases bone density in mice. An inhibitor to the succinate receptor shows a trend to decrease succinate induced osteoclastogenesis *in vitro*. Metformin is also used to show a decrease in succinate in the MKR mice which display an increase in bone density. In addition, bone marrow cells obtained from mice treated with metformin show decreased osteoclastogenesis when treated with succinate *in vitro*. The studies provide a variety of approaches to address the role of succinate in osteoclastogenesis. However, the link isn't fully demonstrated at this point and further studies are needed. In particular, the studies would greatly benefit from additional cohort studies, blood glucose/body weight and bone density data, additional *in vivo* osteoclast measures and optimally a study showing that blocking succinate receptor *in vivo* prevents bone loss resulting from diabetes or succinate treatment.

Major comments:

1) Type 2 diabetes is not usually associated with bone loss. If type 2 diabetes is associated with elevated succinate levels which increase osteoclastogenesis, then one wonders how trabecular bone density is maintained. More information/discussion on how this could occur would be helpful in the introduction; this will set a stronger basis for the studies.

Response 1: Thanks for the suggestion. More information/discussion on how this could occur has been added in the introduction. Increased cortical porosity has been observed in T2DM patients which suggests an association of increased cortical bone resorption in diabetic patients.

2) General information on the mice in each experiment is critical for interpreting the results, but not included. It is important to show the mouse blood glucose or HbA1C levels as an indicator of

diabetes and its severity – this is never shown in any of the studies and is an important control. Similarly, mouse weights should be shown for each study since weight can influence bone density. In addition, because there are several mouse experiments, it would be helpful to have the mouse ages in the figure legends.

Response 2: The mouse blood glucose levels, weights and ages have been added (Figure 1L and S1).

3) Information on when succinate increases during diabetes development in the MKR mice would be helpful, if possible. Does it increase immediately with diabetes onset?

Response 3: We have added the succinate levels in different aged MKR mice (under 4-week-old, 6~8-week-old and 12-week-old). The starting of the succinate increase correlates well with the diabetes onset which is around 6~8-week-old (Supplementary Figure S1B-C).

4) Bone density should be included in the first study (Figure 1), since the authors are showing increased osteoclastogenesis. Does this lead to bone loss?

Response 4: Bone density measured by microCT analysis has been included in the revised manuscript (Figure 1G-H). MKR mice exhibit increased osteoclast activity and reduced BMD than WT mice which we have reported by us ¹ and others ².

5) In figure 1 the only measure of activated osteoclasts is histological, additional measures such as serum TRAP would strengthen this data. It is also critical to show that the results reproduce in another cohort of mice; it would be best to increase the n in the second run since an n of 4 seems quite low.

Response 5: Serum TRAP5b level is included (Fig. 1K). We agree that the number of mice is not high. However, the group size has provided enough statistic power and confirmed the previous reports on increased osteoclast activity in MKR mice reported by us ¹ and others ².

6) Figure 1 legend is wrong – 1C legend discusses BMSCs from WT and MKR mice, but they show serum succinate levels.

Response 6: We corrected the mistaken legend.

7) Figure 2B: When doing t-test, what is being compared? Are all conditions compared? When counting TRAP pos cells, is the final number per well? Per view? Per mm²?

Response 7: The t-test was compared to PBS treated group after ANOVA. The counting was per view. We added these information to the legend in this revised manuscript.

8) Figure 2 – given that the serum contains around 100um of succinate, it would be useful to quantitate the osteoclast numbers in the succinate dose response which ranges from 100um (in diabetic serum) to 1mM (which is used in all in vitro studies).

Response 8: We observe a moderate but not significant effect with 100um succinate in osteoclastogenesis (Figure 2C and S2A). We believe that the succinate released from BMSCs provides a gradient of succinate levels which could yield a much higher local succinate concentration around osteoclast precursors adjacent to BMSCs than the succinate level detected in the serum.

9) Figure 3 – Is the succinate solution endotoxin free? Succinate may increase inflammation leading to osteoclast activation. It would be helpful to use the antagonist or a receptor KO to prove that succinate is directly working through the receptor.

Response 9: All the reagents including water used for solutions are cell culture grade which should be endotoxin free. We have conducted new experiment with SUCNR KO mouse bone marrow cultures and proved that succinate stimulating osteoclastogenesis requires the receptor (Figure 4D-E).

10) Figure 3 – similar to past comments, the data would be strengthened by additional measures of osteoclast activity (ie, serum markers) as well as including an additional cohort of animals that repeats the findings.

Response 10: We used up all the serum left from the old cohort of mice in cytokine ELISA assays (Figure 3L-M). So we confirmed the effects of succinate with a new cohort of WT mice and measured cytokines and the osteoclast activity with serum marker TRAP5b levels (Figure S4G). The results supported our original observation that succinate stimulates osteoclast activity *in vivo*.

11) Figure 3 – Were other cytokines measured? What were their levels? Can the measured cytokines be given a concentration? It is unclear what the “density” means.

Response 11: The “density” from a cytokine array was similar to Western blot which is determined by the intensity of the binding on the membrane. We have measured more osteoclastic cytokines including IL-1beta and TNFalpha using conventional ELISA kits; both IL-1beta and TNFalpha were significantly upregulated in mice treated with succinate (Figure 3L-M and Figure S4H-I).

12) Figure 4 – The receptor antagonist did not significantly decrease TRAP MNCs (#/???) in figure 4C. Also, in figure 4E succinate did not significantly increase TRAP MNCs, suggesting variability between repeat studies. Statistics need to use a post-hoc anova.

Response 12: As demonstrated in the Figure 4 in this revised manuscript, we have repeated the 4c *in vitro* experiment and this time we seeded more cells initially and the the antagonist effects became more significant: it blocked succinate stimulation on osteoclastogenesis (Figure 4B-C). Meanwhile, we added results from SUCNR1 KO and siRNA cultures which supported the effect of succinate via SUCNR1 (Please also refer our Response 3 to Reviewer 1).

13) It would be helpful if a rationale was given for the treatment periods, ie: why 14 days treatment? Were other shorter time points tried that did not have an effect?

Response 13: I assume the reviewer meant 14 days treatment with metformin. We didn't try less than 14-days treatment considering the lifespan of osteoclasts *in vivo* is several weeks and 14-days treatment could be the shortest time to have an effect.

14) Figure 5 – Again, there aren't any mouse parameters such as blood glucose, body weight. Metformin could affect hyperglycemia and the succinate changes may be secondary to this.

Response 14: The levels of blood glucose and body weight in metformin versus PBS treated mice have been added (Figure 5Q-R).

15) What about using WT mice that are treated with succinate and then giving these mice metformin? This would remove the hyperglycemia component.

Response 15: The focus of this study is on succinate stimulating osteoclastogenesis through its receptor which has been demonstrated by succinate administration *in vitro* and in WT (no hyperglycemia component) *in vivo*. The reduction of succinate in diabetic mice by metformin is likely through re-activation of SDH, therefore, we don't anticipate that metformin could reduce the external administered succinate level in mice.

16) Figure 6 – Blood glucose levels of treated mice?? This study was a bit confusing since the mice were treated with metformin and then bone marrow stromal cells isolated and then treated with succinate. Figure 6C should be noted as WT OC cultures? It isn't fully clear what is going on in this figure.

Response 16: We apologize for the confusion. This experiment was conducted with bone marrow cells derived from PBS or metformin treated mice. Fig. 6C is in MKR OC cultures. More details have been added to the legend.

Reviewer #3 (Remarks to the Author):

The paper reports that succinate enhances osteoclast formation, most likely through activation of the previously orphan GPCR, now known to be the succinate receptor (SUCNR1). They used MKR mice which rapidly develop hyperglycaemia due to type II diabetes. They recently published data obtained with the use of metabolomics, showing in bone marrow, elevated levels of TCA cycle metabolites, particularly succinate, which is the object of this study.

Part of this paper confirms the recent publication (reference 13), showing the great increases in succinate produced by BMSCs from hyperglycaemic animals. The present paper also they show greatly elevated plasma levels of succinate to the order of 100 μ M, as well as decreased bone with increase in osteoclast parameters.

In view of the known increase in osteoclasts and resorption in diabetic bone disease they investigated the possible role of succinate. There were some known findings that seemed to make this worthwhile. For example, succinate through SUCNR1 was known to promote NF κ B

translocation, to increase intracellular Ca in an IP-dependent manner via PLCbeta, and to inhibit PGE-increased cAMP (ref 15).

The rationale of the work is clearly explained, but it suffers much from poorly explained methods and experiments, among other problems.

Specific matters.

1. Figure 2, several questions.

In the text and in several figure legends, the term "primary OC cultures" is used. This is wrong and misleads. These are primary BM cultures treated with RANKL and M-CSF to generate osteoclasts over several days, and that should be made clear in every instance.

Indeed the description of relevant method is deficient. BM cells are removed and plated out etc, but there is no statement of the size of culture wells or the cell density. This is particularly important in these experiments because BM cultures of a certain density will have sufficient accompanying stromal cells to mediate differentiation of osteoclasts through RANKL production with appropriate treatment. Here it is just not clear. They appear to treat with RANKL and M-CSF in all experiments, but it is not clear whether they need it in all – they never state what the cell density etc is, and particularly, as I refer to below, in all their experiments the actual number of osteoclast generated appears very low – what were the size of the wells in the density of cultures etc? The data it is therefore extremely difficult to interpret without further details.

Response 1: We have corrected the term and added more details including the plate size and cell density in each experiment. RANKL and M-CSF were added in all *in vitro* bone marrow cell culture experiments. Unless otherwise stated, the cell seeding numbers were 20,000/well in 96-well-plate.

2. Figure 3. Were these bone data obtained with n = 5 per group? This is a small number for such experiments particularly when such small effects were obtained. Were the animals pair-fed? Any effect of succinate on feeding and hence on growth over the seven weeks?

Fig 3K. I am assuming that this is the result of OC measures in bone sections. If it is *ex vivo* generation from narrow, exact details would be required. If it is *in vivo* data it is given as "TRAP+ve/mm". More important is a measure of OC surface.

Why select IL-4 and IL-17 from among the many cytokines that either inhibit or increase osteoclast formation? This is not interpretable, quite apart from the fact that the IL-4 data is not significant.

Response 2: The size of group has been added in each experiment. In addition, we have repeated the *in vivo* succinate treatment experiments in a new cohort of WT mice (Figure S4) and confirmed the effect of succinate.

No significant effect of succinate on the body weight was observed in both cohorts (Figure 3N and Figure S4).

We re-measured the surface of the OCs in all TRAP staining slides, and now reported the results on osteoclast surface per bone surface (Oc.S/mm) (Figure 1J, 3J, 5H, and S3D).

The cytokine array we used were not able to detect changes in other cytokines. The serum samples were diluted to cover the entire membrane during incubation, we didn't detect strong signals of other cytokines. Here, we conducted new experiments using individual cytokine

ELISA kits in two cohorts of succinate treated mice. We found the osteoclastogenic cytokines such as IL-1beta and TNFalpha were significantly upregulated in mice treated with succinate (Figure 3L-M and Figure S4H-I).

3. 4B, C. 1mM succinate was used to treat the primary marrow cultures (? Also treated with RANKL, M-CSF?? Not stated). Again, the number of OCs generated is very small indeed. Importantly though, if 10 UM of a SUCNR1 antagonist is considered to inhibit specifically a 1mM concentration of agonist, this would require some pharmacological explanation, as it would be quite remarkable.

4D. In this experiment they are comparing the numbers of osteoclasts generated by ex vivo BM cultures from MKR mice with or without SUCNR1 antagonist treatment in vivo. Again there are very few OCs generated and a small effect claimed. Comparison of the responses in the control versus MKR requires a 2-tailed ANOVA to compare responses, and not just a comparison of the levels obtaining with the two treatments as the authors have done.

Besides, such an experiment carried out on cells in vitro needs very careful equivalence of starting numbers of BM cells in the culture. Methods do not go into this at all.

Response 3: RANKL and M-CSF were always added. We did experience some variations with the sizes and numbers of bone marrow cultures may be due to the nature of the culture includes mixed cell population, the different batches of FBS and mice. But the effects of succinate has been very robust regardless of these factors.

The affinity of small compound 4c as SUCNR1 antagonist could be much higher than succinate as a natural ligand. Normally, SUCNR1 is not activated by succinate at its physiology level which is supported by the overall healthy phenotype of SUCNR1 KO mice. Therefore, succinate to SUCNR1 affinity is not high while 4c was selected to antagonize SUCNR1 to block SUCNR1 at much lower concentration.

One of the key findings of our study is succinate acts through SUCNR1 to exert its effect in osteoclastogenesis. We have confirmed this with siRNA and SUCNR1 KO OC cultures. In this revised manuscript, we have repeated the experiment of 4c *in vitro* administration and received similar result. We have performed ANOVA analysis before t-test. The detailed information about cell concentrations and plate sizes have been added to Methods and legends. The initial cell numbers were always equal among groups.

4. Fig 5. Metformin appeared to reduce OC formation in vitro of MKR mice cells by about 20%, but appeared to have a greater effect on resorption in vitro. This method approach is seriously questioned. According to methods, the resorption was measured at the end of a period in which OCs were generated on the surface to be resorbed. Therefore the findings at the end of culture was actually dependent on the numbers of osteoclasts generated. The usual way such a question is addressed is to generate the osteoclasts, and transpose equal numbers of them onto bone or dentine slices and measure their ability to resorb.

Contributing to the discrepancy in this case might also be the artificial resorption substrate used, which, unlike dentine or bone can "resorb" in response to macrophages and leucocytes.

Response 4: We agree and as pointed out by the reviewer the Osteo-Surface may yield some "artificial resorption" *in vitro*. Importantly, the effect of succinate or metformin on osteoclast activity has been confirmed by *in vivo* experiments (Figure 3, Figure S4, and Figure 5).

5. There is available a mouse with globally knocked out SUCNR, that is viable and healthy, and could be used to get much nearer to a definitive answer to their main research question.

Response 5: Thanks for the suggestion. We recently obtained the KO breeders and have conducted experiment with bone marrow cells derived from SUCNR KO mice. The result support that succinate stimulates osteoclastogenesis via the receptor (Figure 4D-C).

References:

- 1 Li, X., Guo, Y., Yan, W., Snyder, M. P. & Li, X. Metformin Improves Diabetic Bone Health by Re-Balancing Catabolism and Nitrogen Disposal. *PloS one* **10**, e0146152, doi:10.1371/journal.pone.0146152 (2015).
- 2 Kawashima, Y. *et al.* Type 2 diabetic mice demonstrate slender long bones with increased fragility secondary to increased osteoclastogenesis. *Bone* **44**, 648-655, doi:10.1016/j.bone.2008.12.012 (2009).

Reviewers' comments:

Reviewer #1 (Remarks to the Author):

The authors have dealt with my concerns and I am happy to recommend acceptance.

Reviewer #3 (Remarks to the Author):

The revised paper leaves a number of questions unanswered, in addition to other problems.

1. The question raised by the effect of a 10 μ M concentration of the low molecular weight compounds inhibiting a 1mM concentration of agonist is not adequately addressed. The specificity of this finding is not established, there is no dose response, and no attempt to investigate the effects on responses to other treatments that enhance a RANKL effect, e.g, TGFbeta (e.g. JBMR 6:1787, 2001). The latter specificity test is missing also with the shRNA experiments added in revision.

2. There is no information providing any dose-dependence of the succinate effect on RANKL - induced osteoclast formation. Neither the osteoclastogenesis data in Fig 2 or Fig S2 provide quantitative dose-dependence – just a single high dose of 1mM, and selected microscopic fields chosen to claim dose dependence. That is not an acceptable approach. The magnitude of the in vitro effect is not great, and the in vivo response could be indirect (v infra). In the case of the TGF enhancement of RANKL-induced osteoclast formation, that response is greater than that shown here. Furthermore, when hemopoietic cells are co-cultured with osteoblastic cells to generate osteoclasts with various stimuli, TGFbeta actually inhibits osteoclast formation (JBMR 6:1787, 2001). That level of complexity of these responses has not been addressed in this work.

3. Further, there is still no credible suggested mechanism by which succinate might enhance the RANKL response. Enhancement of RANKL-induced osteoclast formation is a well-established effect of TGFbeta. At least that possibility might have been investigated. Mechanistic possibilities could also have been investigated out of their new data shown in revision, that succinate increased production in vivo of two selected cytokines, IL-1 and TNFalpha. As with TGFbeta, these could be among any of a number of mediators of an indirect action of succinate. They and even other cytokines could contribute tom the in vivo effects that are noted. None of these possibilities have been explored.

4. The authors persist in calling the cultures "primary osteoclast cultures", which is incorrect and misleading. These are cultures of marrow precursors that are extremely heterogeneous, and only quite a low percentage of the cells in any of these cultures eventually become osteoclasts with RANKL and M-CSF treatment.

5. An inadequate response is also provided to the criticism of their claim that succinate increased osteoclast resorptive activity as well as their formation. They generated the osteoclasts in the presence of treatment, and measured resorption on the same plates. Therefore the amount of resorption was determined by the number of osteoclasts formed. The same claim was made, using this method, with regard to the metformin treatment, where the claim is that metformin has a greater effect on resorption than on osteoclast formation. The authors refer to the in vivo data in Figs 3I, S4 and 5 in support of their in vitro findings and conclusions. These Figs show increased numbers of active osteoclasts on bone surfaces. That data in no way answers the criticism of their interpretation of the in vitro data. The only way changes in osteoclast resorption activity can be shown is to generate the osteoclasts, remove them and settle them in equal numbers of cells on fresh plates , and measure the resorption. This fundamental error is made commonly as a result of lack of understanding of methods in osteoclast biology. It results in presentation of misleading and potentially confusing conclusions.

We thank the first two reviewers for recommending acceptance of our manuscript. We have conducted new experiments and revised our manuscript according to the reviewer 3' comments.

Reviewers' comments

Reviewer #3

1. The question raised by the effect of a 10 μ M concentration of the low molecular weight compounds inhibiting a 1mM concentration of agonist is not adequately addressed. The specificity of this finding is not established, there is no dose response, and no attempt to investigate the effects on responses to other treatments that enhance a RANKL effect, e.g, TGFbeta (e.g. JBMR 6:1787, 2001). The latter specificity test is missing also with the shRNA experiments added in revision.

Response 1: We have conducted new experiments including dose responses of 4c (new Figure 4B-C) and siRNA (New Figure S5D-E), both 4c and SUCNR1-siRNA demonstrated dose-dependent inhibitory effects on osteoclastogenesis.

2. There is no information providing any dose dependence of the succinate effect on RANKL induced osteoclast formation. Neither the osteoclastogenesis data in Fig 2 or Fig S2 provide quantitative dose dependence – just a single high dose of 1mM, and selected microscopic fields chosen to claim dose dependence. That is not an acceptable approach. The magnitude of the in vitro effect is not great, and the in vivo response could be indirect (v infra). In the case of the TGF enhancement of RANKL induced osteoclast formation, that response is greater than that shown here. Furthermore, when hemopoietic cells are cocultured with osteoblastic cells to generate osteoclasts with various stimuli, TGFbeta actually inhibits osteoclast formation (JBMR 6:1787, 2001). That level of complexity of these responses has not been addressed in this work.

Response 2: We have quantitatively assessed the succinate dose response experiment (New Figure 2C-D). The focus of this manuscript is on a novel effect of succinate in osteoclastogenesis directly through its receptor SUCNR1. Our study does indicate that succinate may indirectly stimulate osteoclastogenesis through cytokines such as IL-1beta and TNFalpha (Figure 3L-M and Figure S4H-I), but the detailed mechanism requires further study which is beyond the scope of this manuscript.

3. Further, there is still no credible suggested mechanism by which succinate might enhance the RANKL response. Enhancement of RANKL induced osteoclast formation is a well established effect of TGFbeta. At least that possibility might have been investigated. Mechanistic possibilities could also have been investigated out of their new data shown in revision, that succinate increased production in vivo of two selected cytokines, IL1 and TNFalpha. As with TGFbeta, these could be among any of a number of mediators of an indirect action of succinate. They and even other cytokines could contribute to the in vivo effects that are noted. None of these possibilities have been explored.

Response 3: Our study suggest that succinate may enhance RANKL stimulated osteoclastogenesis via NF-kB signaling. We observed enhanced p65 and p50 nuclear localization in WT but not in SUCNR1 KO cells using indirect Immunofluorescence (New Figure

6D) and Western blotting (New Figure S6A). It is not clear at this time whether succinate/SUCNR1 activation interact with TGFbeta. We will explore that possibility in our future study.

4. The authors persist in calling the cultures "primary osteoclast cultures", which is incorrect and misleading. These are cultures of marrow precursors that are extremely heterogeneous, and only quite a low percentage of the cells in any of these cultures eventually become osteoclasts with RANKL and MCSF treatment.

Response 4: We removed the term "primary osteoclast" across this revised manuscript. All the osteoclast cultures now are referred as "osteoclasts derived from mouse bone marrow cells".

5. An inadequate response is also provided to the criticism of their claim that succinate increased osteoclast resorptive activity as well as their formation. They generated the osteoclasts in the presence of treatment, and measured resorption on the same plates. Therefore the amount of resorption was determined by the number of osteoclasts formed. The same claim was made, using this method, with regard to the metformin treatment, where the claim is that metformin has a greater effect on resorption than on osteoclast formation. The authors refer to the in vivo data in Figs 3I, S4 and 5 in support of their in vitro findings and conclusions. These Figs show increased numbers of active osteoclasts on bone surfaces. That data in no way answers the criticism of their interpretation of the in vitro data. The only way changes in osteoclast resorption activity can be shown is to generate the osteoclasts, remove them and settle them in equal numbers of cells on fresh plates, and measure the resorption. This fundamental error is made commonly as a result of lack of understanding of methods in osteoclast biology. It results in presentation of misleading and potentially confusing conclusions.

Response 5: We modified our statement on the osteoclast resorptive activity and bone resorption considering that we only observed increased numbers of active osteoclasts on bone surface and a reduction in bone volume in succinate treated mice in vivo. We now conclude that succinate plays a role in stimulating osteoclastogenesis.

REVIEWERS' COMMENTS:

Reviewer #3 (Remarks to the Author):

The authors have responded to the 5 points made in my last review, and providing dose response data for the succinate effect on osteoclast formation and for the inhibitor effect.

In the case of Fig 5 however, although the authors have modified their claim that succinate stimulates osteoclast activity in addition to stimulating osteoclastogenesis, they have retained a sentence that persists with this claim:

Lines 133-134: "The OC activity was also increased by succinate since the area of the resorption pits was significantly increased by 3-fold (Figure 2E-F)."

It is clear from Fig 2D that 1mM succinate increases osteoclast formation 5-fold. The resorption data is obtained from those cells growing on the same plates, and the resorbed area is 3-fold greater with 1 mM succinate(Fig 2F) – thus explained entirely by the increased osteoclast numbers. There is no evidence of a separate effect of succinate on osteoclast activity, distinct from the effect on formation, as pointed out in my last review. The authors have not addressed the question of a separate effect on osteoclast activity. They are two different processes and that claim should not be retained. The sentence in lines 133-134 should be removed.

We have revised our manuscript according to the reviewer 3' comments. Please see below:

Reviewer #3 comments

The authors have responded to the 5 points made in my last review, and providing dose response data for the succinate effect on osteoclast formation and for the inhibitor effect.

In the case of Fig 5 however, although the authors have modified their claim that succinate stimulates osteoclast activity in addition to stimulating osteoclastogenesis, they have retained a sentence that persists with this claim:

Lines 133-134: "The OC activity was also increased by succinate since the area of the resorption pits was significantly increased by 3-fold (Figure 2E–F)."

It is clear from Fig 2D that 1mM succinate increases osteoclast formation 5-fold. The resorption data is obtained from those cells growing on the same plates, and the resorbed area is 3-fold greater with 1 mM succinate (Fig 2F) – thus explained entirely by the increased osteoclast numbers. There is no evidence of a separate effect of succinate on osteoclast activity, distinct from the effect on formation, as pointed out in my last review. The authors have not addressed the question of a separate effect on osteoclast activity. They are two different processes and that claim should not be retained. The sentence in lines 133-134 should be removed.

Response: We understand the concern and removed the lines 133-134.